**Subject Category:**
Biology (whole organism)

ecology

provisioning, habitat use, behaviour, dispersal kernel, cryptic function loss, seed retention time

**Author for correspondence:**
Joanna K. Carpenter
e-mail: carpenterj@landcareresearch.co.nz

# Long seed dispersal distances by an inquisitive flightless rail (*Gallirallus australis*) are reduced by interaction with humans

Joanna K. Carpenter[1,2], Colin F. J. O'Donnell[3], Elena Moltchanova[4] and Dave Kelly[2]

[1]Manaaki Whenua-Landcare Research, Private Bag 1930, Dunedin, New Zealand
[2]Centre for Integrative Ecology, School of Biological Sciences, University of Canterbury, Private Bag 4800, Christchurch 8140, New Zealand
[3]Department of Conservation, Biodiversity Group, Christchurch, New Zealand
[4]Department of Math and Statistics, University of Canterbury, Christchurch 4800, New Zealand

JKC, 0000-0001-7666-2439; DK, 0000-0002-9469-2161

Human presence is becoming increasingly ubiquitous, but the influence this has on the seed dispersal services performed by frugivorous animals is largely unknown. The New Zealand weka (*Gallirallus australis*) is an inquisitive flightless rail that frequently congregates in areas of high human use. Weka are important seed dispersers, yet the seed dispersal services they provide are still poorly understood. We estimated seed dispersal distances of weka for two plant species (*Prumnopitys ferruginea* and *Elaeocarpus dentatus*) and tested how human interaction affected these dispersal distances. We estimated weka seed dispersal distances by combining GPS data from 39 weka over three sites with weka seed retention time data in a mechanistic model. The mean seed retention times were extremely long (38–125 h). Weka were highly effective dispersers, dispersing 93–96% of seeds away from parent canopies, and 1% of seeds over 1 km. However, we found evidence of a significant human impact on the seed dispersal distances of weka, with birds occupying areas of high human use performing 34.8–40.9% shorter distances than their more remote counterparts. This represents an example of cryptic function loss, where although weka are still present in the ecosystem, their seed dispersal services are impaired by human interaction.

# 1. Introduction

Humans have modified over half of the earth's terrestrial surface [1], with profound consequences for the species that use those habitats. Landscape modification consists of structural changes such as roads or deforestation, and non-structural changes such as noise pollution or frequent human activity [2]. Unsurprisingly, animals in these anthropogenically disturbed habitats move and behave differently to their undisturbed counterparts. Most studies report decreased vagility of animals living in modified habitats, driven by both negative effects such as barriers to movement (e.g. [3]) and positive effects such as enhanced resources at modified sites (e.g. crops, supplementary feeding and water sources) that mean animals have to travel less to meet their resource requirements (e.g. [4]). Due to these mechanisms, movements of mammals in intensively human-modified areas can be reduced by half to two-thirds compared with individuals in areas with less human modification [5], and similar reductions in vagility have been documented for other taxa (e.g. [6,7]).

Reduced animal movements could affect important ecosystem functions such as seed dispersal. The movement patterns of seed-dispersing animals have large effects on how far seeds travel, the individual survival prospects of seeds and the probability of long-distance dispersal events [8]. Seed deposition patterns then affect recruitment and colonization rates, the ability of plants to escape climate change effects, gene flow between plant populations, and forest community composition [9–12]. All these problems are magnified because at the same time as fragmentation reduces the mobility of seed dispersers, it increases the dispersal distances required to maintain gene flow and may reduce the density of dispersers. In fragmented landscapes, maintaining gene flow between patches of remaining habitat requires longer dispersal distances, but reduced vagility of dispersers makes that less likely [13]. For example, carnivorous mammals moved seeds shorter distances in fragmented forests compared with more intact habitats [14], potentially resulting in reduced gene flow between forest fragments. In addition, many seed disperser guilds in modified habitats have greatly declined in both abundance and species diversity, so the maintenance of seed dispersal services may rely on a fraction of the species that it once did [15,16]. Therefore, anthropogenic landscapes may induce cascading effects for plant communities by truncating the movements of seed dispersers.

Even in landscapes that do not suffer from structural anthropogenic modifications like roads or habitat fragmentation, human presence is becoming increasingly pervasive, and this may directly affect the vagility and effectiveness of seed-dispersing animals. For example, mantled howler monkeys (*Alouatta palliata mexicana*) inhabiting a forest fragment used for nature-based tourism spent more time in lower quality habitat when the number of human visitors increased [17]. Similarly, rhesus macaques have shortened daily ranges when they are fed by people, suggesting reduced seed dispersal distances [18]. However, few studies have investigated how human presence may alter the movement and effectiveness of seed-dispersing animals, despite the fact that even 'wild' landscapes are becoming increasingly crowded with people.

In New Zealand, one species that has a high level of human interaction is the weka (*Gallirallus australis*; Sparrman, 1786), an inquisitive flightless rail (figure 1*a*). Weka (as a Maori word, the spelling is the same for singular and plural) are bold, opportunistic birds that frequently aggregate at areas of high human use, such as campsites or picnic areas [19]. Their charisma and cunning results in them often being fed by, and stealing food from, people [19]. Recently, weka have been shown to be important seed dispersers for some New Zealand plant species [20], but they have been studied much less than volant, frugivorous birds such as the kereru (New Zealand wood pigeon *Hemiphaga novaeseelandiae* [21]). As a result, no one has measured how long weka retain seeds in their guts, and how far they disperse seeds. In addition, understanding the dispersal distances provided by weka may give insights into the seed dispersal capabilities of flightless rails as a group, which were once common across the Pacific but have suffered widespread extinctions there in the last 3000 years [22].

We used a mechanistic model to study weka seed dispersal distances. Mechanistic models combine data on seed retention times and high-resolution animal movement patterns to estimate the magnitude and frequency of potential seed dispersal distances [23]. Our aims were to: (i) estimate weka seed retention times using a novel radio-frequency identification (RFID) microchip method, which records when ingested microchipped seeds were still inside weka; (ii) estimate weka seed dispersal distances for two large-seeded plant species by collecting weka movement data and combining these with seed retention data in a mechanistic model; and (iii) investigate how interaction with humans affects weka seed dispersal distances. We predicted that weka that spent a lot of time in areas of high human use would move less far, and therefore disperse seeds shorter distances, than their counterparts in more remote areas.

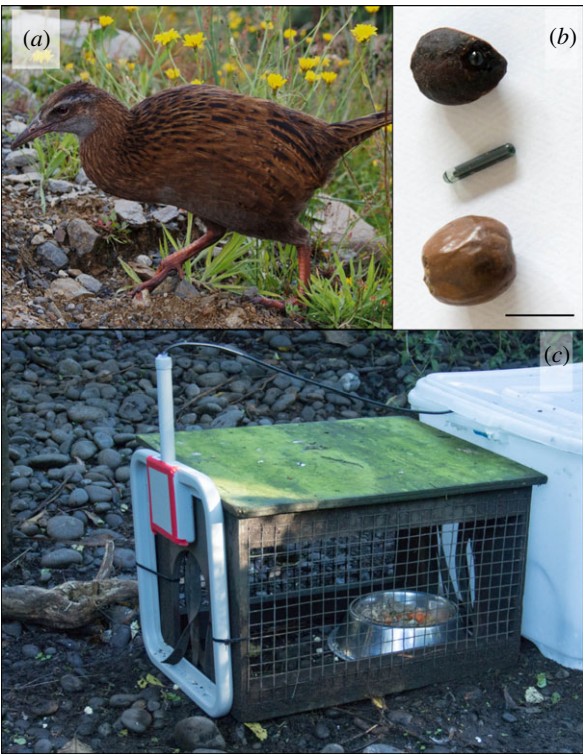

**Figure 1.** Use of PIT tags to estimate weka (*Gallirallus australis*) seed retention times. (*a*) Weka. (*b*) Miro (*P. ferruginea*, top) seed with PIT tag inserted inside before epoxy resin was applied; 12.5 mm long PIT tag; hinau (*E. dentatus*, bottom) seed with PIT tag inserted inside. Scale bar is 10 mm. (*c*) Photo showing the RFID antenna (grey) which weka had to move through to access their food.

## 2. Methods

### 2.1. Study species

Weka are one of the largest (mean 900 g, range 400–1700 g) extant frugivorous birds in New Zealand [24]. They occur across most habitat types, although their abundance and range have decreased alarmingly since human arrival, due to habitat loss, predation by exotic mammals and drought-related starvation [24]. Their wild diet is dominated by fruit and invertebrates but also includes lizards, carrion, and the eggs and chicks of ground-nesting birds [24]. They swallow fruits whole and either defecate or regurgitate the seeds intact [20,25].

We modelled seed dispersal distances for two large-seeded plant species commonly eaten by weka—miro (*Prumnopitys ferruginea*, Podocarpaceae) and hinau (*Elaeocarpus dentatus*, Elaeocarpaceae) [20,21]. We were particularly interested in these two species, as weka have been shown to be the most important disperser for hinau in terms of fruit removal [20], and they are also likely to be an important disperser for miro where weka are present, as its large fruits are consumed by few other species [21]. In addition, both plant species feature some fruit traits (e.g. early abscission, thick seed coats) that suggest they may be partially adapted for dispersal by flightless birds such as weka [26], and are found throughout all (miro) or nearly all (hinau) of the range of weka. Miro is a tree that grows to 25 m tall and occurs throughout New Zealand. Its fruits are 12–15 mm in diameter, with a fleshy exocarp and a hard, woody seedcoat 1.5–2 mm thick that encases the single seed [27]. Hinau trees grow up to 20 m tall and occur in lowland conifer-broadleaf forest throughout the North and South Islands. Its fruits average 9.2 mm diameter, with a carbohydrate-rich exocarp and mesocarp and a hard woody seedcoat protecting the single seed [27].

### 2.2. Seed retention times

We collected miro and hinau seed retention time data using four captive weka (three females and one male) held in two enclosures (250 and 110 m² in size) at Willowbank Wildlife Reserve (43°46′ S,

172°59′ E) in Christchurch. While most seed retention time studies are conducted by simply feeding captive birds fruits and observing them until they defecate the seeds, a pilot study with weka indicated that their seed retention times were probably too long to make this method feasible. Therefore, we developed a novel method of measuring long seed retention times using passive integrated transponder (PIT) tagged seeds. A PIT tag is a glass-encapsulated microchip programmed with a unique alphanumeric code. An RFID receiver–transmitter attached to an antenna reads the code remotely. Feeding weka PIT-tagged seeds then regularly scanning the birds until the PIT-tagged seeds were no longer detected allowed better estimates of long gut passage times without the need for continuous observation or searches of enclosures for seeds. We collected miro fruit in May 2018 from Ulva Island (46°92′ S, 168°12′ E), and hinau fruit in June 2018 from the University of Canterbury campus (43°52′ S, 172°58′ E). We drilled a $13 \times 2.5$ mm hole into the long axis of each seed. We first removed the exocarp from the miro seeds to prevent the seed slipping out of the vice while drilling. A single glass-encapsulated high-frequency PIT tag ($12 \times 2.15$ mm, weight 0.1 g, Oregon RFID) was glued into the hole with epoxy resin (figure 1*b*). We did not weigh the seeds before and after modification, but as the seed material we removed was replaced by the PIT tag, any change in weight would be minor.

We offered the captive weka PIT-tagged seeds at various times of day, and recorded the time that each PIT-tagged seed was ingested. PIT-tagged seeds were smeared with cheese to make them more attractive to the birds. We used a Chafon 13.56 MHz RFID ISO15693 middle range reader and a Chafon 13.56 MHz ABS handheld high-frequency antenna ($352 \times 332 \times 22$ mm) to detect PIT-tagged seeds inside the weka. The square antenna was fixed to the door of the weka's food shelter, so weka walked through the antenna to reach their food (figure 1*c*). A laptop attached to the reader logged the identity and time when PIT-tagged seeds were detected by the scanner. This system allowed the simultaneous reading of multiple PIT tags, which is important when birds have multiple seeds in their gut.

We calculated the time that elapsed between each PIT-tagged seed being eaten and the last time it was detected by the scanner. The PIT-tagged seeds were easily detected inside the birds' guts by the scanner antenna, with each PIT-tagged seed being detected an average of 10.6 times per day. As PIT tag detectability is affected by the angle that the PIT tag is on and whether birds passed completely through the antenna, not all PIT-tagged seeds were detected every time the birds passed through the scanner antenna. However, each PIT-tagged seed was detected 79.7% of the time on average. There was no apparent variation among individual birds in the timing of their activity. Therefore, the seed retention times we estimated based on the last detection time in the gut were likely to be only small underestimates. We recorded weka consuming some PIT-tagged seeds that were never subsequently detected by the scanner, but this was rare (1 out of 19 seeds for miro, and 3 out of 21 hinau seeds). In these cases, we believe the bird regurgitated the seeds before passing through the scanner. We therefore used the first scan where other seeds were detected since feeding the missing seeds as an indicator of their retention time.

During the duration of the trial, weka were kept on their regular diet of commercially available fruit and vegetables (apples, peas, etc.), commercial pet food and seeds. Weka were fed at mid-morning each day. After the trials ended, we searched one of the enclosures for the seeds that had experienced the longest retention times to check whether they were intact. As seeds were extremely difficult to find visually (due to the substrate in the enclosure), we used the RFID antenna (attached to the scanner and laptop) to scan the ground, working in a sweeping motion.

## 2.3. Collection of GPS data for weka seed dispersal kernels

We used three sites to collect high-resolution movement data for weka in order to estimate weka seed dispersal distances (figure 2). Lake Mahinapua Scenic Reserve (42°79′ S, 170°90′ E) and Goldsborough Reserve (42°67′ S, 171°12′ E) are two areas of dense podocarp-broadleaf forest near Hokitika on the West Coast of the South Island. The third site, Ulva Island, is a 267 ha island off Rakiura/Stewart Island, which is free of exotic mammalian predators (e.g. rats *Rattus* spp.), and covered in podocarp-broadleaf forest.

At two of these sites, we could also assess how human presence affects weka seed dispersal distances. Lake Mahinapua and Goldsborough both contain popular Department of Conservation campsites, which are areas of high human use. These campsites are surrounded by dense podocarp-broadleaf forest, which is rarely entered by people. This allowed us to sample weka that contact humans frequently (those that spent time in the campsite) versus infrequently (those in the forest away from the campsite). Ulva Island had no hub of high human use so was not used for this analysis, but was still used to estimate overall weka seed dispersal distances.

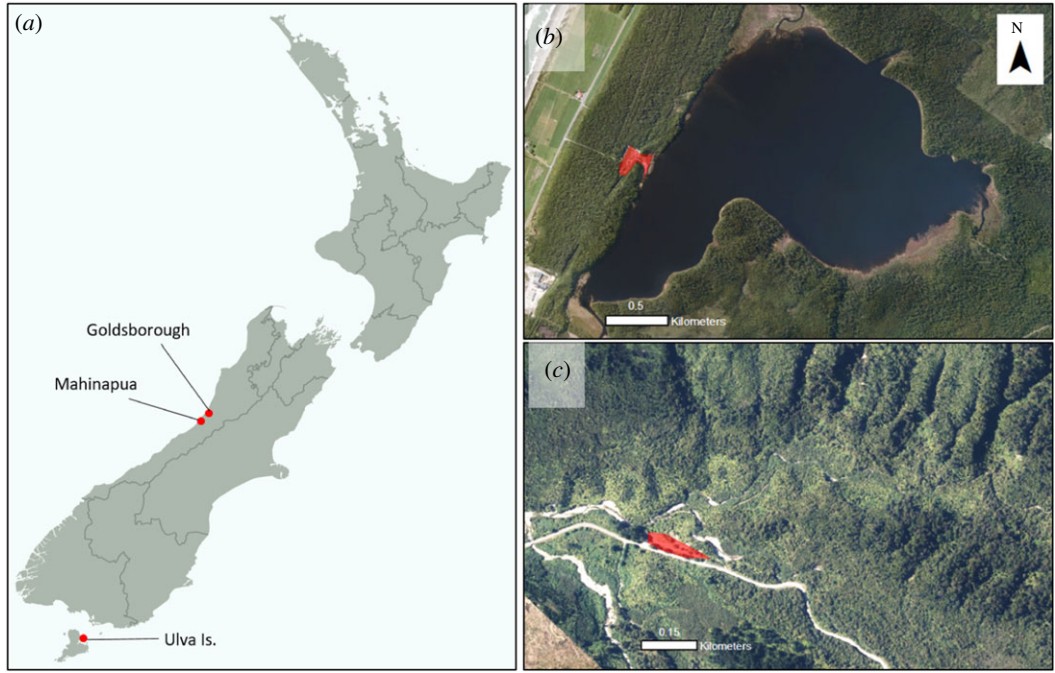

**Figure 2.** Weka GPS tracking locations. (*a*) Map of New Zealand showing the location of the three sites. (*b,c*) Satellite images of the sites at Lake Mahinapua and Goldsborough, respectively, with campsite areas shaded red.

Collection of movement data occurred between February and May 2018 (the peak of the fruiting season for most bird-dispersed New Zealand plant species [27]). We captured 46 weka using either a ground noose [28] or hand net, and took weight and bill measurements from each individual. Where possible, we sexed and aged (juvenile or adult) each bird based on its weight, bill measurements, wing spurs and vocalizations [29]. We used electrical heatshrink plastic to combine into one package an igot-u120 GPS logger and a Sirtrack Ultimate Lite single-stage VHF transmitter (combined weight 30 g), and secured it to the bird using a backpack harness. Birds were only fitted with a GPS and harness if the combined unit was no more than 5% of their body weight, as heavier devices relative to bird body mass can result in negative effects on foraging, locomotion and physiology [30]. GPS tags were programmed to attempt a fix every 15 min. After 14 days (the battery life of the GPS), we recaptured the weka and removed the transmitter and harness. Following difficulties recapturing some of the Lake Mahinapua birds, we fitted birds from Goldsborough and Ulva Island with a harness that included a degradable weaklink [31]. Three weka were unable to be recaptured after the two-week monitoring period (two individuals from Lake Mahinapua and one individual from Goldsborough), and four weka had their GPS devices fail. This left us with GPS data from 39 weka (Lake Mahinapua $n = 13$, Goldsborough $n = 12$, Ulva Island $n = 14$), including both sexes ($F = 11$, $M = 19$; 5 individuals from Ulva Island and 4 juveniles from Goldsborough and Mahinapua could not be sexed) and juvenile ($n = 4$) and adult individuals ($n = 35$).

We excluded from analysis all GPS fixes that were obtained using fewer than four satellites due to concerns about their lack of accuracy [32]. We also visually inspected the waypoints for each bird and removed any waypoints that appeared implausible (i.e. waypoints that occurred so far away from the last recoded waypoint that the bird could not have moved there in that time, and single waypoints that occurred large distances away from the bird's usual home range with no waypoints in between). The final dataset retained 66% of the original fixes. Previous studies have shown that the GPS tags we used have a mean location error of less than 10 m, even under dense cover [32].

## 2.4. Mechanistic model

We developed a mechanistic seed dispersal model using the GPS data from all three sites and the weka seed retention times to estimate the distribution of weka dispersal distances (dispersal kernels) for miro and hinau. Seed retention times were simulated by randomly sampling 100 000 seed retention times from the weka seed retention time data that we had collected for the two plant species. We removed the two

longest hinau seed retention datapoints (30 and 40 days) as they were considerably longer than the 14 day GPS logs. That means that our estimated dispersal distances were biased towards shorter seed retention times.

Weka can forage at all times of the day and night ([26]; J.K.C. 2017, unpublished data), so we simulated ingestion for all hours of the day. The model simulated ingestion of each seed at randomly selected times during the bird's tracking log. As GPS waypoints were obtained at 15 min intervals, when the model selected a sample time between two waypoints, we interpolated the location of the bird (drew a line between the last recorded waypoint and the next recorded waypoint and calculated the location of the bird between those two fixes, assuming that the bird was moving in a linear direction at a constant speed). The model then used the randomly sampled seed retention time to determine the time of defecation, found the location of the bird at that time and measured the distance between the ingestion and defecation locations. The process was repeated to obtain a probability distribution of dispersal distances for each bird, which were then pooled to obtain the mean dispersal distance for each plant species.

When the simulated seed retention time ran past the end of the tracking session, the resulting dispersal distance was recorded as missing. This means that dispersal distances were further biased towards shorter seed retention times. For miro, an average of 5% of dispersal distances were missing (range 0–20% for different birds) because the track stopped before the seed emerged, and for hinau, an average of 4.25% dispersal distances were missing (range 0–31%).

## 2.5. Analysis

In order to identify the birds that had a high level of human interaction (Lake Mahinapua and Goldsborough birds only), we used bivariate normal kernel functions to estimate the utilization distribution of each bird's home range [33]. Individuals whose core home range (defined as the 70% isopleth [28]) overlapped with the campground were defined as birds that had a high level of human interaction (hereafter referred to as 'camp followers'). Birds at the two sites whose core home range did not overlap with a campground were categorized as 'remote' birds. We only used adult birds for the analysis as juvenile weka have different movement patterns to adult birds [34] and all the juvenile birds we captured were camp followers ($n = 4$), which would have confounded the model. Our analyses were therefore performed on 10 adult camp follower birds, and 11 adult remote birds. We used a linear mixed effects model to assess whether camp followers had shorter mean dispersal distances than remote birds. The median dispersal distance for each bird for each plant species (obtained from the mechanistic model) was the response variable, while human interaction (or not), site and plant species were the fixed effects. We did not include interaction terms for the fixed effects as the model without interactions had the lowest AIC value. We used the median dispersal distance for each bird because we were interested in variability between individuals rather than within individuals. We log-transformed the median dispersal distance for each bird to improve normality. We used Levene's tests to confirm homogeneity of variances. Sex was initially included in the models as a fixed effect but it was non-significant, so we removed it. Bird ID was included as a random effect. All analyses were conducted in R v. 3.5.1, using the packages lme4, lmerTest, adehabitatHR and car.

# 3. Results

## 3.1. Weka seed retention times

The four captive weka consumed 19 PIT-tagged miro seeds and 21 PIT-tagged hinau seeds, with each bird consuming between 4 and 15 seeds. Three of the birds consumed seeds from both species. The weka had very long seed retention times (figure 3). The mean for miro was 38.5 h (s.d. = 85), with a range of 1–379 h (up to ca 15 days) and a median of 8.2 h. The mean retention time for hinau was 125.2 h (s.d. = 252.2), with a range of 2.5–958 h (up to ca 40 days) and a median of 20.5 h. Hinau seed retention times did not differ from miro seed retention times (Kruskal–Wallis test, $\chi^2 = 2.6$, d.f. = 1, $p = 0.10$).

Two weeks after the trials finished, we relocated seven seeds from the largest enclosure using the antenna and scanner, including the two seeds that had been inside the weka for 14 and 15 days. All of the seeds were intact.

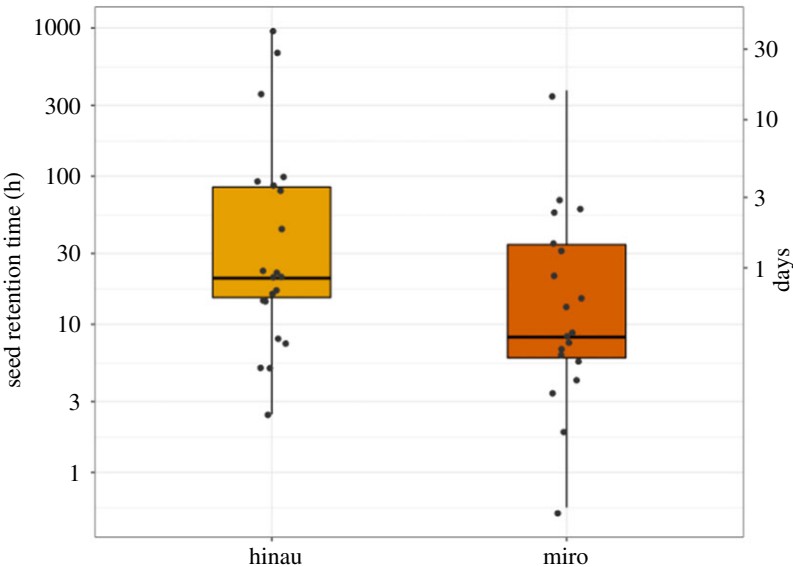

**Figure 3.** Distribution of weka gut retention times (log scale) for seeds of hinau (*E. dentatus*, n = 21) and miro (*P. ferruginea*, n = 19). The grey dots show the actual data points, while the boxplot shows median, interquartile range, minimum and maximum.

**Table 1.** Weka seed dispersal distances (m) and distribution patterns (% in distance bands) generated from the mechanistic model for miro (*P. ferruginea*) and hinau (*E. dentatus*).

| | dispersal distance (m) | | seeds dispersed (%) | | | |
|---|---|---|---|---|---|---|
| plant species | mean (±1 s.d.) | maximum | <10 m | 10–100 m | 100–1000 m | >1000 m |
| miro | 125.5 (±175.3) | 2333 | 7 | 54 | 38.2 | 0.8 |
| hinau | 142.8 (±197.9) | 2113 | 4 | 52 | 42.9 | 1.1 |

### 3.2. Weka seed dispersal kernels

The mechanistic model estimated that weka dispersed 93–96% of seeds away from the parent tree (assuming a canopy radius of 10 m; table 1). Just over half of the seeds were dispersed within 100 m of the source, approximately 40% of seeds were dispersed over 100 m, and around 1% of seeds were dispersed over 1 km. The dispersal kernels appeared to have a leptokurtic (long-tailed) distribution for both plant species (figure 4). Miro and hinau mean dispersal distances were broadly similar (table 1).

### 3.3. The effect of human interaction on mean seed dispersal distances by weka

Our linear mixed effect model demonstrated that camp follower weka dispersed hinau and miro seeds significantly shorter distances than remote birds (table 2). We estimated the mean dispersal distances for camp followers and remote birds (using the median dispersal distance for each individual bird) and found that miro seeds were dispersed 53.4% further by remote birds (mean ± s.e. miro dispersal distance = 113.5 ± 20.2 m) compared with camp followers (74 ± 11.5 m) (figure 5). Hinau seeds were dispersed 69.3% further by remote birds (mean + s.e. hinau dispersal distance = 138.8 ± 27.4 m) compared with camp followers (81.9 ± 12.7 m) (figure 5).

## 4. Discussion

### 4.1. Weka seed retention times

Our seed retention time trials indicated that weka, a flightless omnivorous rail, have the longest avian seed retention times yet recorded. To our knowledge, emus (*Dromaius novaehollandiae*), least sandpipers (*Calidris minutilla*) and killdeers (*Charadrius vociferus*) are the only avian species with

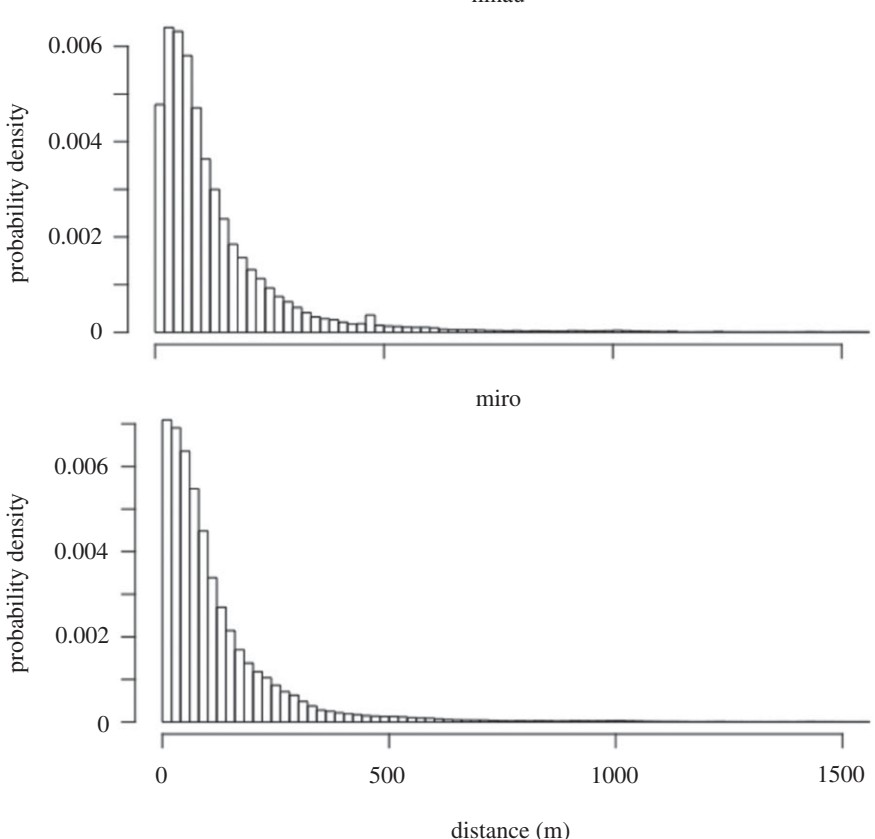

**Figure 4.** Probability distributions of weka seed dispersal distances at 20 m intervals for miro (*P. ferruginea*) and hinau (*E. dentatus*), generated by a mechanistic model using empirical data for individual weka movements and seed retention times.

**Table 2.** Outputs of linear mixed effects model of dispersal distances conducted by weka with high human interaction (camp followers) and low human interaction (remote birds) for miro (*P. ferruginea*) and hinau (*E. dentatus*). Human interaction is a significant fixed effect.

| random effects | variance | s.d. | | |
|---|---|---|---|---|
| individual | 0.20 | 0.44 | | |
| fixed effects | estimate | s.d. | *t*-value | *p*-value |
| (intercept) | 4.10 | 0.17 | 23.56 | <0.001*** |
| low human interaction (remote birds) | 0.45 | 0.20 | 2.30 | 0.033* |
| plant species (miro) | −0.14 | 0.03 | −4.70 | <0.001*** |
| site | 0.48 | 0.20 | 2.42 | 0.026* |

\* $p < 0.05$, \*\*\* $p < 0.001$.

comparable reported seed retention times to weka [35,36]. The only quantitative study on emu seed retention times using actual seeds (rather than artificial seed mimics) found that approximately 90% of seeds were recovered within 24 h, with a maximum retention time of 264 h [37], which are nearly as long as weka. Killdeers and least sandpipers had maximum seed retention times of 340 and 216 h, respectively [36], but the mean seed retention times were not reported. In comparison, most volant passerines have seed retention times of less than 60 min [38–40].

The long seed retention times of weka are probably due to their relatively large size, flightlessness and diet. There is a positive relationship between bird size and seed retention times [41–43]. Flightlessness probably further lengthens seed retention times as flightless birds have no evolutionary selection to aid flight by having short intestines that quickly remove heavy seed 'ballast' [44], and can thus afford to retain gut contents for longer for more complete digestion. Diet and digestive strategy

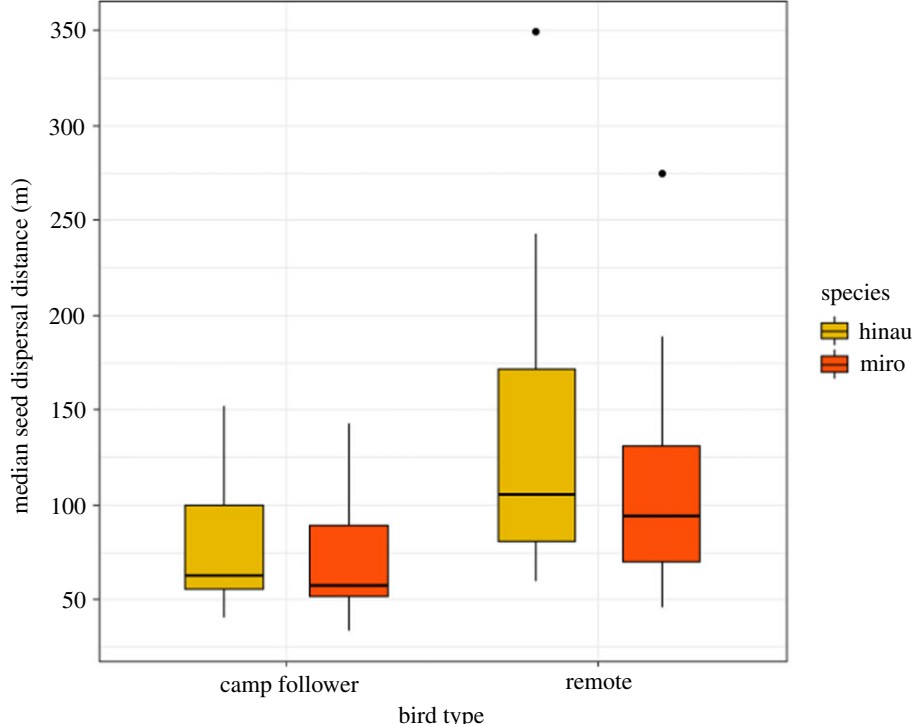

**Figure 5.** Modelled dispersal distances for hinau (*E. dentatus*) and miro (*P. ferruginea*) for 'camp follower' weka that experienced a lot of human interaction (*n* = 10), and 'remote' weka that had very little human interaction (*n* = 11). Linear mixed effects models showed the differences were significant for both species; table 2.

are also strong predictors of seed retention time in birds, with specialist frugivores having shorter seed retention times than herbivores or omnivores [43]. Weka may have slow gut passage in order to effectively process invertebrates, animal proteins and coarse vegetable matter [45,46]. In addition, our removal of the fruit pulp of miro may have affected weka seed retention times for that species, as the secondary metabolites within fruit pulp can decrease or increase seed retention times [47–49]. Here, we found long retention times both in hinau (which had the fruit pulp intact) and miro (which had the fruit pulp removed), although we cannot determine what might have happened if fruit pulp had been removed for hinau or left intact for miro. Similarly, the cheese we smeared on the fruits may have affected the weka seed retention times, although we think this is unlikely, given how long the seeds remained inside the birds.

Interestingly, weka retain grit and small stones in their gizzard to help process food, and it seems likely that the seeds which remained inside the weka for greater than 14 days were acting as substitute gizzard stones. Five of New Zealand's tree species, including miro and hinau, have unusually thick, woody seed coats which have previously been identified as 'anachronistic', or maladapted for dispersal by the contemporary fauna [21]. Our finding that weka retain some of these seeds for longer than two weeks offers a possible explanation for these species' unusually thick seed coats (relative to other New Zealand trees). Consistent with this hypothesis, the recovered seeds, which spent two weeks in the gut, had polished seed coats but appeared otherwise undamaged. However, it should be noted that long seed retention times can have trade-offs between increased seed dispersal distances and decreased germination of seeds [50]. Assessing the germination success of our weka-passed seeds was not possible as inserting PIT tags damages the seeds, but anecdotal evidence suggests that seeds germinate well after weka gut passage, with seed-filled weka droppings being used to germinate seedlings for plant nurseries (G. Davidson 2018, personal communication). Carpenter *et al*. [26] also demonstrated that mild abrasion improved the germination of hinau seeds, so the mild abrasion that occurs when seeds pass through the grit-filled gizzard of weka could be beneficial for certain plant species.

## 4.2. Weka seed dispersal kernels

Weka dispersal kernels appeared to be leptokurtic in shape, with just over half the seeds predicted to be deposited within 100 m of the source tree, followed by a rapid decline and a long tail out beyond 1000 m.

This distribution is typical for animal-mediated seed dispersal kernels [11]. The weka dispersal kernels were similar to seed dispersal kernels calculated for medium-sized volant birds dispersing mahaleb cherry (*Prunus mahaleb*) in Spain [51], and for spotless starling (*Sturnus unicolor*) and song thrush (*Turdus philomelos*) dispersing wild olive (*Olea europaea* var. *sylvestris*) [52]. Because weka dispersal distances were large relative to canopy area, remarkably few seeds (4–7%) would be deposited by weka beneath parent canopies. Since seeds that are deposited beneath parent canopies can suffer from disproportionate mortality due to density- and distance-dependent mortality [53], this result demonstrates that weka provide highly effective seed dispersal [54]. As a comparison, the kereru, New Zealand's largest volant seed disperser, is relatively sedentary and disperses 13–34% of seeds beneath parent canopies [42], so weka provide better seed dispersal than kereru in this respect.

Despite being flightless, the mean dispersal distances of weka compare favourably with dispersal distances calculated for some of New Zealand's volant frugivores, which are also capable of consuming hinau or miro (although it should be noted that two common frugivores, bellbirds (*Anthornis melanura*) and tui (*Prosthemadera novaeseelandiae*) have not been recorded consuming hinau). Weka mean dispersal distances are greater than those calculated for kereru using a similar mechanistic model (61–98 m [42]). They are probably also greater than bellbird dispersal distances, given bellbird's small home range size (0.02 ha [55]), but this is still untested. However, weka mean seed dispersal distances are smaller than those estimated for tui, a highly mobile passerine (214–231 m [56]), although the maximum dispersal distances calculated for tui (2.2 km) are similar to weka. These results suggest that weka contribute non-redundantly to seed dispersal kernels by providing a complementary service to other New Zealand frugivores.

While weka seed dispersal distances are probably constrained by their home range sizes, their extremely long seed retention times mean that they have considerable potential to occasionally move seeds very long distances. As we only recorded weka movements for 14 days, we had a limited capacity for recording any rare long-distance movements, although some juvenile weka moved over 1–2 km away from their usual range. Interestingly, some adult weka on Ulva Island also moved 1–2 km away from their usual home ranges, perhaps to take advantage of food resources on beaches. Although flightless, weka are proficient swimmers and easily cross environmental barriers such as major rivers, lakes and mountain ranges [57]. Coleman *et al.* [57] recorded an adult male weka moving 35 km away from their study site, with one juvenile moving 9 km away, and translocated weka have been recorded moving 600 km over a six-week period. These occasional long-distance movements, coupled with seed retention times that can reach 40 days, demonstrate that weka almost certainly generate rare long-distance dispersal events beyond what could be documented with our mechanistic model (which also underestimated dispersal distances for several technical reasons as noted in Methods). These long-distance dispersal events could profoundly enhance genetic flow across extensive landscape scales, as well as helping plants track their climatic envelope and colonize new sites.

## 4.3. The influence of humans on weka seed dispersal distances

We found that weka that spent a large amount of time at places of high human use had shorter estimated dispersal distances than their more remote counterparts. The differences in modelled dispersal distances between camp followers and remote birds were large, with remote birds dispersing seeds 53–69% further than camp followers. Reduced dispersal distances could easily influence the likelihood of seeds reaching suitable microsites. For example, in fragmented habitats, a truncation of seed dispersal distances could be enough to reduce the number of seeds dispersed to suitable habitat fragments, thereby reducing gene flow between plant populations (e.g. [58]). However, it should be noted that in some cases, human disturbance may have the potential to increase seed dispersal distances of some species, for example, when water birds are regularly disturbed and fly to different wetlands [59]. In addition, it is possible that there were other unmeasured differences associated with the campsites (other than human presence) which affected weka seed dispersal distances.

While we did not investigate the mechanisms driving this trend in weka, provisioning by humans is probably important in shortening the seed dispersal distances of camp follower birds. Weka that scavenge calorie-rich food at campsites or are deliberately fed by people (a common sight at campsites and picnic areas) would have to move less far to meet their energy requirements than birds that solely eat a wild diet. Several other studies have found that animals move less when they are provisioned. Supplementary feeding decreased the home range size of red deer (*Cervus elaphus*) in Slovenia [60], and anthropogenic food resources led to smaller home ranges of raccoons (*Procyon lotor*) in Illinois, USA [61]. Provisioning by humans is also likely to lead to decreased levels of frugivory in weka,

although we did not measure whether human-habituated weka consumed less wild fruit. Similarly, Sengupta *et al*. [18] demonstrated that provisioned rhesus macaques had decreased frugivory and seed dispersal activities.

Our results provide an example of how the increasing ubiquity of people across the globe may have hidden consequences for seed dispersal processes, although the impacts of human presence on seed disperser behaviour will differ depending on the mechanisms involved. This finding is an example of cryptic function loss—where the ecological function of an animal population is significantly altered as a result of anthropogenic disturbance, even though the species is still present in the ecosystem [2] and in fact is paradoxically increased in visibility by the same processes that simultaneously decrease its effectiveness for dispersal services.

## 4.4. Conservation implications in New Zealand and elsewhere

Weka are important, yet unappreciated, seed dispersers in New Zealand [20]. They have been recorded consuming the fruits of over 26 native fleshy-fruited plant species, including some of New Zealand's largest-seeded species [62], and the fruits of low-growing divaricating shrubs. Weka may also disperse the seeds of monocots or dry seeds through granivory, an overlooked yet important seed dispersal mechanism that has been demonstrated for other rail species such as Eurasian Coots (*Fulica atra*) [63]. Our study has shown that weka also provide highly effective seed dispersal services, by dispersing 93–96% of seeds away from parent canopies—a higher proportion than kereru, which is often called New Zealand's most important seed disperser (discussed in [21]). In addition, weka contribute to gene flow between plant populations by dispersing a reasonable proportion of seeds beyond 1 km.

Unfortunately, weka are controversial for New Zealand conservation managers because of the predatory impacts they have on other native fauna. This has led to them being frequently excluded from restoration projects [19], even though they have been lost from large areas of their native range, and are still threatened by exotic mammalian predators. We urge conservation managers to consider the positive contributions weka can also make to ecosystem functioning when debating their presence in restoration projects. In addition, greater attempts should be made to discourage people from feeding weka.

More broadly, our research into the seed dispersal capabilities of weka suggests that other rails across the Pacific may have been important seed dispersers, even though they are rarely mentioned as such. However, *Porphyrio* species have been recorded as significant vectors of *Coprosma* seeds both within and between New Zealand and the Pacific, and *Scleria* seeds were found in the gut of *Porphyrio* in Fiji (cited in [64]). Steadman [22] estimates at least 450 rail species have gone extinct across the Pacific in the last 3000 years, and the ecological consequences of those losses are still unknown. Even if just a small fraction of those species had the seed dispersal capabilities of weka, then the loss of Pacific rails may represent one of the most widespread yet least appreciated losses of dispersal function ever recorded.

Ethics. All fieldwork carried out was conducted with approval from the New Zealand Department of Conservation, under C.F.J.O'D's. research program. All applicable institutional and/or national guidelines for the care and use of animals were followed. This work was carried out with University of Canterbury Animal Ethics Committee Approval (2017/32R and 2017/22R).

Data accessibility. All data used in this paper are available as electronic supplementary material, S1 and S2. The code used to run the mechanistic model is available as electronic supplementary material, S3.

Authors' contributions. J.K.C. and D.K. designed the study, J.K.C. collected the data with logistical assistance from C.F.J.O'D. and D.K., J.K.C. and E.M. analysed the data, J.K.C. wrote the manuscript and all authors provided editorial advice.

Competing interests. The authors state they have no conflict of interest.

Funding. This work was funded by grants from Koiata Trust, Birds New Zealand and the Brian Mason Scientific and Technical Trust. J.K.C. was supported by a University of Canterbury Roper Scholarship in Science and a Graduate Women New Zealand Award.

Acknowledgements. We thank Theo Thompson, Jane Furkett, Blair Sandford, Archie MacFarlane, Robyn Long, Kim Roberts and Robyn White for fieldwork assistance, and the helpful staff at Willowbank Wildlife Reserve. Joris Tinnemans provided training in weka capture, Susan Waugh provided advice on GPS tags, and Ralph Powlesland, Graeme Taylor and Jim Briskie gave advice during the planning stages. Paul van Klink and Jim Watts provided excellent advice on capturing weka, and Stuart Cockburn from the Department of Conservation provided invaluable technical support. Thanks also to the Department of Conservation staff at Hokitika and Rakiura who made this work possible, and to the Hunter family for allowing access to their lands on Ulva Island. We thank Geoff Walls for highlighting the mutualistic relationship between weka and hinau.

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
