## [Reviewer comments · Royal Society Open Science]

Review History

RSOS-190397.R0 (Original submission)

Review form: Reviewer 1

Is the manuscript scientifically sound in its present form?

Yes

Are the interpretations and conclusions justified by the results?

Yes

Is the language acceptable?

Yes

Is it clear how to access all supporting data?

No

Do you have any ethical concerns with this paper?

No

Have you any concerns about statistical analyses in this paper?

Yes

Recommendation?

Accept with minor revision (please list in comments)

Comments to the Author(s)

Manuscript ID: RSOS-190397

GENERAL COMMENTS:

The manuscript entitled 'Long seed dispersal distances by an inquisitive flightless rail (*Gallirallus australis*) are reduced by interaction with humans' is an original work through which authors combine GPS data and gut retention time to estimate seed dispersal distances provided by the New Zealand weka, and evaluate the impact that a strong contact with humans can have on this service. The study has involved a major field effort. It provides interesting results on cryptic function loss and opens debate on conservation strategies.

The manuscript is clearly written, easy to follow and concise. However, I have some comments that I hope they help to improve the manuscript.

-ABSTRACT:

l 38-40: I think that reporting the number of individuals analysed is not relevant in the abstract. Maybe you could rewrite this part of the abstract: 'However, we found evidence of a significant human impact on the seed dispersal distances of weka, birds occupying areas of high human use showing 34.8-40.9% shorter distances than their more remote counterparts'.

- KEYWORDS: I suggest replacing 'habitat change' by 'habitat use', which I think fits better to the studied situation in which structural anthropogenic modifications of the landscape are not described.

- INTRODUCTION:

l. 54: add a comma after 'noise'

l. 61-64: your focal seed disperser species is a bird, so why don't you change this example by highlighting what has been reported in birds (ref. 6)?

l. 71. This is a very general statement about seed dispersal consequences, so I think that ref. 9 is not necessary here (it refers to a certain plant species). But if you want to keep it, it should be necessary to fix it in the reference list (author names are not correctly written and two of them are repeated).

l. 96. I suggest adding the authorship of the species the first time you mention it in the manuscript. The same for other taxa.

l. 104. Carpenter 2018 unpubl. data: Why don't you include these data on gut retention time as part of this study?

l. 110. ...to study weka seed dispersal distances.

METHODS

l. 124. birds (plural)

l. 130-138: Any reference supporting these descriptions of the plant species?

l. 139 onwards: The study has an important spatial component, so it would be nice to see a figure with a map showing the three sites studied and the area occupied by the campsites at the Mahinapua and Goldsborough Reserves. Besides, the current manuscript has only one figure, so maybe a first figure showing graphically the experimental design could be part of the body of the ms.

l. 166. Justify why the device shouldn't be heavier than 5% of the bird's body mass and add a

reference (e.g. Geen et al 2019. Effects of tracking devices on individual birds – a review of the evidence. *Journal of Avian Biology*, e01823)

l. 167. Why 14 days of monitoring? I guess because of the lifespan of the GPS batteries, but justify why.

l. 170. Were these three birds those from Mahinapua Reserve?

l. 173. Report n of males and females, and n of juvenile and adults.

l. 175. Report the % of GPS fixes excluded.

l. 183 onwards (Mechanistic model). As required by the journal: “Datasets, code, and other digital materials should be deposited in an appropriate, recognised, publicly available repository”.

Therefore I recommend you to make the R-code for the Mechanistic model available as supporting information.

l. 189. As pointed above, I think that this study focused on the estimation of seed dispersal distances by combining GPS data and information about gut retention time could be the place where properly report the data you have on seed retention time.

l. 196. Delete ‘only’

l. 218-225. It is not clear from the text if n=10 camp followers includes the juveniles or not, because they are also camp followers according to their home range. Rewrite this part in order to clarify that juvenile birds are n=5 and the final analysis was performed with 21 adult birds.

l. 235. You included ‘site’ as a random effect in your linear mixed effect model. However, you are only considering 2 sites, and the use of random factors with less than 5-6 levels is not recommended. Crawley [p. 670: Crawley, M. J. 2002. *Statistical Computing: An Introduction to Data Analysis using S-PLUS*] recommended treating variables as fixed factors when there aren’t enough levels of a factor.

As published by Bolker et al. in the following blog:

<http://bbolker.github.io/mixedmodels-misc/glmmFAQ.html#should-i-treat-factor-xxx-as-fixed-or-random>

Are there enough levels of the factor in the data on which to base an estimate of the variance of the population of effects? No, means [you should probably treat the variable as] fixed effects.

- RESULTS

l. 243. This info (< 7%) is redundant. You already said that 93-96% of the seeds are dispersed away from the parent tree.

l. 253. Cite Table 2 instead of Table 1.

DISCUSSION

l. 268 You cite an important study on spatial patterns of seed dispersal, but I miss here other important and more recent studies reporting seed dispersal kernels by birds of different sizes to which it would be interesting compare and discuss your dispersal kernel:

e.g.

Jordano et al. 2007. Differential contribution of frugivores to complex seed dispersal patterns. *PNAS*, 104

González-Varo et al. 2017. Unravelling seed dispersal through fragmented landscapes: Frugivore species operate unevenly as mobile links. *Molecular Ecology*, DOI: 10.1111/mec.14181

l. 272. I would be more careful when talking about effectiveness in seed dispersal (also in l. 345), since it involves many other drivers besides seed dispersal distance. I think that your results are very important for the qualitative component of the seed dispersal effectiveness. I also but think that you can talk about effectiveness when discussing your results together with previous information on the quantitative component of seed dispersal.

l. 284-286. But are these other frugivores capable of dispersing the seeds of the large-fruited Hinau and Miro? Please, rewrite to clarify that weka and these other frugivores overlap in their

fruit diet so they could provide complementary seed dispersal services.

l. 292. Do you know if juveniles perform dispersal movements in a certain time of the year? If so, could these movements involve long distance dispersal events for certain plant species with overlapping fruit phenologies? (This is just a question that rises to me when reading this line).

l. 304-305. Not only on enhancing gene-flow but also for colonising new sites and tracking the climatic envelope.

l. 317. Add references to support this.

Review form: Reviewer 2

Is the manuscript scientifically sound in its present form?

No

Are the interpretations and conclusions justified by the results?

No

Is the language acceptable?

Yes

Is it clear how to access all supporting data?

No

Do you have any ethical concerns with this paper?

No

Have you any concerns about statistical analyses in this paper?

Yes

Recommendation?

Major revision is needed (please make suggestions in comments)

Comments to the Author(s)

See the attached file (Appendix A).

Review form: Reviewer 3

Is the manuscript scientifically sound in its present form?

Yes

Are the interpretations and conclusions justified by the results?

Yes

Is the language acceptable?

Yes

Is it clear how to access all supporting data?

No

Do you have any ethical concerns with this paper?

No

Have you any concerns about statistical analyses in this paper?

No

Recommendation?

Major revision is needed (please make suggestions in comments)

Comments to the Author(s)

Carpenter et al. explore the efficiency of flightless birds, in particular the New Zealand rail weka, as seed dispersers, and further investigate the effects of increased exposure to human presence in the landscape. Using GPS tracks from some 40 birds in three different sites combined with mechanistic modeling of seed retention time, they find that while volant birds are usually presumed more efficient seed dispersers, the flightless weka is a surprisingly efficient disperser due to relatively large home ranges and unusually long seed retention time. However, wekas are curious birds that are attracted to human activity, and the authors noticed reduced mobility among rails in areas with human presence (e.g. campgrounds), leading to cryptic loss of ecosystem services through reduced seed dispersal.

This is overall a neat and well-designed study that delivers both counterintuitive and important results. I have unusually few remarks, but I do find one issue generally problematic: The mechanistic model of seed retention is based on an unpublished dataset of seed retention times by the main author ("Carpenter 2018 unpubl. data"), from which the statement also stems that the weka has the longest seed retention time in the world. Since this is one of the two major datasets that the present paper is based upon, I do not think it is acceptable to only refer to this as an unpublished dataset (and not make it available). In the present paper, the probability distributions of seed dispersal distances are presented, but as reader I am only presented with a couple of descriptive statistics of the seed retention time. Perhaps this dataset is meant to be published in a separate manuscript, but if that manuscript is not accepted, it renders the present one premature. In order to avoid that, I would argue that the authors should include methods and results for the collection of that dataset too in this manuscript. I also note that the authors have answered "Yes" to the question whether new data is presented (true), and state that "All data used in this paper are available as ESM 1". The GPS logging dataset in ESM 1 is extensive and well curated, but it only covers the tracking, and the latter statement is thus incorrect. Thus, in order to adhere to RSOS standards, the seed retention data should make up ESM 2, and be presented in the manuscript.

I don't think that the opening statement is correct. The authors claim that "Humans have modified approximately 50 to 70% of the earth's surface" and refer to Barnosky et al. (2012) *Nature*. However, I do not believe that these numbers are supported by Barnosky et al. (especially look at terrestrial vs. marine parts of the Earth, and current vs. projected transformation).

The authors consistently use "weka" as plural form. As far as I, as non-newzeelander, understand, "wekas" would be the correct plural form of weka, although I also understand that "weka" is sometimes used also as plural.

References 5, 24, 26, 38, 42 have missing or incorrect details or incorrect formatting. As a very minor comment, I would advise the authors to present Figure 2 as boxplots rather than a bar chart, given the type of data.

Apart from the treatment of seed retention data, all other comments are minor, and would render an "Accept with minor revision". In my opinion, if the availability and treatment of seed retention data (preferably by inclusion in the manuscript and as supplement or data deposition), this manuscript is in good shape and qualifies for publication in RSOS. However, as is, I therefore recommend a "Major revision".

Decision letter (RSOS-190397.R0)

13-Jun-2019

Dear Ms Carpenter,

The editors assigned to your paper ("Long seed dispersal distances by an inquisitive flightless rail (*Gallirallus australis*) are reduced by interaction with humans") have now received comments from reviewers. We would like you to revise your paper in accordance with the referee and Associate Editor suggestions which can be found below (not including confidential reports to the Editor). Please note this decision does not guarantee eventual acceptance.

Please submit a copy of your revised paper before 06-Jul-2019. Please note that the revision deadline will expire at 00.00am on this date. If we do not hear from you within this time then it will be assumed that the paper has been withdrawn. In exceptional circumstances, extensions may be possible if agreed with the Editorial Office in advance. We do not allow multiple rounds of revision so we urge you to make every effort to fully address all of the comments at this stage. If deemed necessary by the Editors, your manuscript will be sent back to one or more of the original reviewers for assessment. If the original reviewers are not available, we may invite new reviewers.

- Data accessibility

<http://datadryad.org/submit?journalID=RSOS&manu=RSOS-190397>

- Competing interests

- Authors' contributions

- Acknowledgements

- Funding statement

Kind regards,

Alice Power

Editorial Coordinator

on behalf of Kevin Padian (Subject Editor)
openscience@royalsociety.org

Associate Editor's comments:

The reviewers of your paper are broadly positive towards it, though each indicate a number of modifications are necessary before it may be considered for publication. Importantly, it has been observed that only a portion of your data have been made available to reviewers (and thus also to readers in the event the paper is published). In line with the journal's data access policy, you should make the dataset(s), code, or other digital research materials available at submission - we will not be able to accept the paper unless you can not only respond effectively to the reviewers' concerns but also provide the additional (and currently absent) data. Please contact the editorial office with any queries you may have before you resubmit.

Comments to Author:

Reviewers' Comments to Author:

Reviewer: 1

Comments to the Author(s)

Manuscript ID: RSOS-190397

GENERAL COMMENTS:

The manuscript entitled 'Long seed dispersal distances by an inquisitive flightless rail (*Gallirallus australis*) are reduced by interaction with humans' is an original work through which authors combine GPS data and gut retention time to estimate seed dispersal distances provided by the New Zealand weka, and evaluate the impact that a strong contact with humans can have on this service. The study has involved a major field effort. It provides interesting results on cryptic function loss and opens debate on conservation strategies.

The manuscript is clearly written, easy to follow and concise. However, I have some comments that I hope they help to improve the manuscript.

-ABSTRACT:

l 38-40: I think that reporting the number of individuals analysed is not relevant in the abstract. Maybe you could rewrite this part of the abstract: 'However, we found evidence of a significant human impact on the seed dispersal distances of weka, birds occupying areas of high human use showing 34.8-40.9% shorter distances than their more remote counterparts'.

- KEYWORDS: I suggest replacing 'habitat change' by 'habitat use', which I think fits better to the studied situation in which structural anthropogenic modifications of the landscape are not described.

- INTRODUCTION:

l. 54: add a comma after 'noise'

l. 61-64: your focal seed disperser species is a bird, so why don't you change this example by highlighting what has been reported in birds (ref. 6)?

l. 71. This is a very general statement about seed dispersal consequences, so I think that ref. 9 is not necessary here (it refers to a certain plant species). But if you want to keep it, it should be necessary to fix it in the reference list (author names are not correctly written and two of them are repeated).

- l. 96. I suggest adding the authorship of the species the first time you mention it in the manuscript. The same for other taxa.
- l. 104. Carpenter 2018 unpubl. data: Why don't you include these data on gut retention time as part of this study?
- l. 110. ...to study weka seed dispersal distances.

METHODS

- l. 124. birds (plural)
- l. 130-138: Any reference supporting these descriptions of the plant species?
- l. 139 onwards: The study has an important spatial component, so It would be nice to see a figure with a map showing the three sites studied and the area occupied by the campsites at the Mahinapua and Goldsborough Reserves. Besides, the current manuscript has only one figure, so maybe a first figure showing graphically the experimental design could be part of the body of the ms.
- l. 166. Justify why the device shouldn't be heavier than 5% of the bird's body mass and add a reference (e.g. Geen et al 2019. Effects of tracking devices on individual birds – a review of the evidence. *Journal of Avian Biology*, e01823)
- l. 167. Why 14 days of monitoring? I guess because of the lifespan of the GPS batteries, but justify why.
- l. 170. Were these three birds those from Mahinapua Reserve?
- l. 173. Report n of males and females, and n of juvenile and adults.
- l. 175. Report the % of GPS fixes excluded.
- l. 183 onwards (Mechanistic model). As required by the journal: "Datasets, code, and other digital materials should be deposited in an appropriate, recognised, publicly available repository". Therefore I recommend you to make the R-code for the Mechanistic model available as supporting information.
- l. 189. As pointed above, I think that this study focused on the estimation of seed dispersal distances by combining GPS data and information about gut retention time could be the place where properly report the data you have on seed retention time.
- l. 196. Delete 'only'
- l. 218-225. It is not clear from the text if n=10 camp followers includes the juveniles or not, because they are also camp followers according to their home range. Rewrite this part in order to clarify that juvenile birds are n=5 and the final analysis was performed with 21 adult birds.
- l. 235. You included 'site' as a random effect in your linear mixed effect model. However, you are only considering 2 sites, and the use of random factors with less than 5-6 levels is not recommended. Crawley [p. 670: Crawley, M. J. 2002. *Statistical Computing: An Introduction to Data Analysis using S-PLUS*] recommended treating variables as fixed factors when there aren't enough levels of a factor.
As published by Bolker et al. in the following blog:
<http://bbolker.github.io/mixedmodels-misc/glmmFAQ.html#should-i-treat-factor-xxx-as-fixed-or-random>
Are there enough levels of the factor in the data on which to base an estimate of the variance of the population of effects? No, means [you should probably treat the variable as] fixed effects.

- RESULTS

- l. 243. This info (< 7%) is redundant. You already said that 93-96% of the seeds are dispersed away from the parent tree.
- l. 253. Cite Table 2 instead of Table 1.

DISCUSSION

- l. 268 You cite an important study on spatial patterns of seed dispersal, but I miss here other important and more recent studies reporting seed dispersal kernels by birds of different sizes to which it would be interesting compare and discuss your dispersal kernel:

e.g.

Jordano et al. 2007. Differential contribution of frugivores to complex seed dispersal patterns. *PNAS*, 104

González-Varo et al. 2017. Unravelling seed dispersal through fragmented landscapes: Frugivore species operate unevenly as mobile links. *Molecular Ecology*, DOI: 10.1111/mec.14181

l. 272. I would be more careful when talking about effectiveness in seed dispersal (also in l. 345), since it involves many other drivers besides seed dispersal distance. I think that your results are very important for the qualitative component of the seed dispersal effectiveness. I also but think that you can talk about effectiveness when discussing your results together with previous information on the quantitative component of seed dispersal.

l. 284-286. But are these other frugivores capable of dispersing the seeds of the large-fruited Hinau and Miro? Please, rewrite to clarify that weka and these other frugivores overlap in their fruit diet so they could provide complementary seed dispersal services.

l. 292. Do you know if juveniles perform dispersal movements in a certain time of the year? If so, could these movements involve long distance dispersal events for certain plant species with overlapping fruit phenologies? (This is just a question that rises to me when reading this line).

l. 304-305. Not only on enhancing gene-flow but also for colonising new sites and tracking the climatic envelope.

l. 317. Add references to support this.

Reviewer: 2

Comments to the Author(s)

See the attached file

Reviewer: 3

Comments to the Author(s)

Carpenter et al. explore the efficiency of flightless birds, in particular the New Zealand rail weka, as seed dispersers, and further investigate the effects of increased exposure to human presence in the landscape. Using GPS tracks from some 40 birds in three different sites combined with mechanistic modeling of seed retention time, they find that while volant birds are usually presumed more efficient seed dispersers, the flightless weka is a surprisingly efficient disperser due to relatively large home ranges and unusually long seed retention time. However, wekas are curious birds that are attracted to human activity, and the authors noticed reduced mobility among rails in areas with human presence (e.g. campgrounds), leading to cryptic loss of ecosystem services through reduced seed dispersal.

This is overall a neat and well-designed study that delivers both counterintuitive and important results. I have unusually few remarks, but I do find one issue generally problematic: The mechanistic model of seed retention is based on an unpublished dataset of seed retention times by the main author ("Carpenter 2018 unpubl. data"), from which the statement also stems that the weka has the longest seed retention time in the world. Since this is one of the two major datasets that the present paper is based upon, I do not think it is acceptable to only refer to this as an unpublished dataset (and not make it available). In the present paper, the probability distributions of seed dispersal distances are presented, but as reader I am only presented with a couple of descriptive statistics of the seed retention time. Perhaps this dataset is meant to be published in a separate manuscript, but if that manuscript is not accepted, it renders the present one premature. In order to avoid that, I would argue that the authors should include methods

and results for the collection of that dataset too in this manuscript. I also note that the authors have answered “Yes” to the question whether new data is presented (true), and state that “All data used in this paper are available as ESM 1”. The GPS logging dataset in ESM 1 is extensive and well curated, but it only covers the tracking, and the latter statement is thus incorrect. Thus, in order to adhere to RSOS standards, the seed retention data should make up ESM 2, and be presented in the manuscript.

I don’t think that the opening statement is correct. The authors claim that “Humans have modified approximately 50 to 70% of the earth’s surface” and refer to Barnosky et al. (2012) Nature. However, I do not believe that these numbers are supported by Barnosky et al. (especially look at terrestrial vs. marine parts of the Earth, and current vs. projected transformation).

The authors consistently use “weka” as plural form. As far as I, as non-newzeelander, understand, “wekas” would be the correct plural form of weka, although I also understand that “weka” is sometimes used also as plural.

References 5, 24, 26, 38, 42 have missing or incorrect details or incorrect formatting.

As a very minor comment, I would advise the authors to present Figure 2 as boxplots rather than a bar chart, given the type of data.

Apart from the treatment of seed retention data, all other comments are minor, and would render an “Accept with minor revision”. In my opinion, if the availability and treatment of seed retention data (preferably by inclusion in the manuscript and as supplement or data deposition), this manuscript is in good shape and qualifies for publication in RSOS. However, as is, I therefore recommend a “Major revision”.

Author's Response to Decision Letter for (RSOS-190397.R0)

See Appendix B.

RSOS-190397.R1 (Revision)

Review form: Reviewer 2

Is the manuscript scientifically sound in its present form?

No

Are the interpretations and conclusions justified by the results?

No

Is the language acceptable?

Yes

Do you have any ethical concerns with this paper?

No

Have you any concerns about statistical analyses in this paper?

Yes

Recommendation?

Accept with minor revision (please list in comments)

Comments to the Author(s)

The paper has clearly improved a lot after full disclosure of the retention time experiment. However, I stand by my earlier comments which the authors have chosen to dismiss. In particular in two sites, birds are compared between two groups, those associated with a camp site and those not. In principle, there could be many reasons why there is a difference in movement between birds using one part of a habitat and another. There are only two places where there is a comparison made between camp and other. Hence it is perfectly possible that the results have a different explanation, the real sample size of importance is $N = 2$ study sites, in both of which a similar spatial pattern is recorded. Given this reality, and in the absence of other data I think the conclusions need to be presented with some more caution.

In this new version I have the following specific comments:

There is a lack of clarity about what is meant by “quality of seed dispersal” e.g. in line 105 or 483. If we compare the paper with Schupp et al. 2017, I do not see what component of quality (as defined in that paper) is being studied herein. The effectiveness framework as presented says very little about dispersal distance, unlike this rail paper. I suggest it is best to just talk of dispersal distance rather than “quality”.

Line 113 Explain/spell out what RFID is.

132-134 A clear justification is required here for why you are modelling dispersal of these two plant species, since you present no data on the consumption or abundance of these two plant species in your study area. Ultimately it is because you have data from a captive experiment, but this needs pointing out. And some idea of the distribution of these two plant species in the study area should be given.

160-161 Discuss the possible consequences of removing the fleshy exocarp from miro seeds on the results, especially retention time. You assume that removing pulp makes no difference, but what studies in other organisms support such an assumption? Furthermore, the seeds were smeared with cheese, might that not influence retention time? At least say why you think not.

433 “very long distance dispersal” seems a significant exaggeration to me for a flightless bird. I would apply that term rather to birds that move a much greater distance of hundreds of km, e.g. Viana et al.

REFERENCES CITED

Schupp, E.W., Jordano, P., Gomez, J.M., 2017. A general framework for effectiveness concepts in mutualisms. *Ecology Letters* 20, 577-590.

Viana, D.S., Santamaria, L., Figuerola, J., 2016. Migratory Birds as Global Dispersal Vectors. *Trends in Ecology & Evolution* 31, 763-775.

Review form: Reviewer 3

Is the manuscript scientifically sound in its present form?

Yes

Are the interpretations and conclusions justified by the results?

Yes

Is the language acceptable?

Yes

Do you have any ethical concerns with this paper?

No

Have you any concerns about statistical analyses in this paper?

No

Recommendation?

Accept with minor revision (please list in comments)

Comments to the Author(s)

I think that the authors have addressed all of the issues brought up by the reviewers in a satisfactory manner, and improved the manuscript greatly. With the explicit inclusion of the seed retention estimation, and the provision of that dataset, the manuscripts is now "complete" and meets the open access standards of RSOS. With a minor edit (see below) I think that this study is now suitable for publication.

After reading the revised manuscript, I only find one minor issue that could easily be addressed. For some reason, in a single passage, only scientific names are used. I would strongly suggest that in the final version of the manuscript, the common names are inserted for lines 395–397. The birds are spotless starling (*Sturnus unicolor*) and song thrush (*Turdus philomelos*), the trees wild olive (*Olea europaea* var. *sylvestris*) and mahaleb cherry (*Prunus mahaleb*) [although the latter one may also be known also under other common names].

Decision letter (RSOS-190397.R1)

29-Jul-2019

Dear Ms Carpenter:

On behalf of the Editors, I am pleased to inform you that your Manuscript RSOS-190397.R1 entitled "Long seed dispersal distances by an inquisitive flightless rail (*Gallirallus australis*) are reduced by interaction with humans" has been accepted for publication in Royal Society Open Science subject to minor revision in accordance with the referee suggestions. Please find the referees' comments at the end of this email.

The reviewers and Subject Editor have recommended publication, but also suggest some minor

revisions to your manuscript. Therefore, I invite you to respond to the comments and revise your manuscript.

- Ethics statement

- Data accessibility

If you wish to submit your supporting data or code to Dryad (<http://datadryad.org/>), or modify your current submission to dryad, please use the following link:
<http://datadryad.org/submit?journalID=RSOS&manu=RSOS-190397.R1>

- Competing interests

- Authors' contributions

- Acknowledgements

- Funding statement

Please note that we cannot publish your manuscript without these end statements included. We have included a screenshot example of the end statements for reference. If you feel that a given

heading is not relevant to your paper, please nevertheless include the heading and explicitly state that it is not relevant to your work.

Because the schedule for publication is very tight, it is a condition of publication that you submit the revised version of your manuscript before 07-Aug-2019. Please note that the revision deadline will expire at 00.00am on this date. If you do not think you will be able to meet this date please let me know immediately.

on behalf of Kevin Padian (Subject Editor)
 openscience@royalsociety.org

Reviewer comments to Author:
 Reviewer: 2

Comments to the Author(s)

The paper has clearly improved a lot after full disclosure of the retention time experiment. However, I stand by my earlier comments which the authors have chosen to dismiss. In particular in two sites, birds are compared between two groups, those associated with a camp site and those not. In principle, there could be many reasons why there is a difference in movement between birds using one part of a habitat and another. There are only two places where there is a comparison made between camp and other. Hence it is perfectly possible that the results have a different explanation, the real sample size of importance is $N = 2$ study sites, in both of which a similar spatial pattern is recorded. Given this reality, and in the absence of other data I think the conclusions need to be presented with some more caution.

In this new version I have the following specific comments:

There is a lack of clarity about what is meant by “quality of seed dispersal” e.g. in line 105 or 483. If we compare the paper with Schupp et al. 2017, I do not see what component of quality (as defined in that paper) is being studied herein. The effectiveness framework as presented says very little about dispersal distance, unlike this rail paper. I suggest it is best to just talk of dispersal distance rather than “quality”.

Line 113 Explain/spell out what RFID is.

132-134 A clear justification is required here for why you are modelling dispersal of these two plant species, since you present no data on the consumption or abundance of these two plant species in your study area. Ultimately it is because you have data from a captive experiment, but this needs pointing out. And some idea of the distribution of these two plant species in the study area should be given.

160-161 Discuss the possible consequences of removing the fleshy exocarp from miro seeds on the results, especially retention time. You assume that removing pulp makes no difference, but what studies in other organisms support such an assumption? Furthermore, the seeds were smeared with cheese, might that not influence retention time? At least say why you think not.

433 “very long distance dispersal” seems a significant exaggeration to me for a flightless bird. I would apply that term rather to birds that move a much greater distance of hundreds of km, e.g. Viana et al.

REFERENCES CITED

Schupp, E.W., Jordano, P., Gomez, J.M., 2017. A general framework for effectiveness concepts in mutualisms. *Ecology Letters* 20, 577-590.

Viana, D.S., Santamaria, L., Figuerola, J., 2016. Migratory Birds as Global Dispersal Vectors. *Trends in Ecology & Evolution* 31, 763-775.

Reviewer: 3

Comments to the Author(s)

I think that the authors have addressed all of the issues brought up by the reviewers in a satisfactory manner, and improved the manuscript greatly. With the explicit inclusion of the seed retention estimation, and the provision of that dataset, the manuscripts is now “complete” and meets the open access standards of RSOS. With a minor edit (see below) I think that this study is now suitable for publication.

After reading the revised manuscript, I only find one minor issue that could easily be addressed. For some reason, in a single passage, only scientific names are used. I would strongly suggest that in the final version of the manuscript, the common names are inserted for lines 395–397. The birds are spotless starling (*Sturnus unicolor*) and song thrush (*Turdus philomelos*), the trees wild olive (*Olea europaea* var. *sylvestris*) and mahaleb cherry (*Prunus mahaleb*) [although the latter one may also be known also under other common names].

Author's Response to Decision Letter for (RSOS-190397.R1)

See Appendix C.

Decision letter (RSOS-190397.R2)

02-Aug-2019

Dear Ms Carpenter,

I am pleased to inform you that your manuscript entitled "Long seed dispersal distances by an inquisitive flightless rail (*Gallirallus australis*) are reduced by interaction with humans" is now accepted for publication in Royal Society Open Science.

on behalf of Prof Kevin Padian (Subject Editor)

Appendix A

This is a study of estimated seed dispersal distances by populations of a flightless rail in three sites. From my understanding, adults and juveniles are tracked at three different sites. Then seed dispersal was estimated only for adults, and only for two of the sites (those with some adult birds using a campsite and other birds using other habitats). Seed dispersal is estimated based on unpublished retention time data, and it seems that neither these data nor the movement data per se are made available by the authors (Q: is that not against the policy of this journal?).

There seems to be no use of real data on true diet or the location or the timing of seed ingestion data, so in that sense the seed dispersal distance estimates cannot be considered to be reliable. Neither are there data presented that show that the rails disperse these plants at these sites, although perhaps some such data are included in reference 21.

The reported effects seem believable and are somewhat equivalent to showing that ducks in urban ponds, where people feed them bread, fly around less (and disperse seeds over shorter distances) than ducks in natural wetlands. This is not very surprising, but there is only a weakly significant effect and the statistical analyses do not seem to me to be robust, since there is really only a sample size of two (i.e. the two locations where birds are compared with and without a campsite). Furthermore, it may be that there is some other confounding variable, e.g. the campsites are positioned in low-lying areas with more emergent vegetation that provides more natural food (if these rails behave anything like *Porphyrio*).

I did not understand how the movement data from the third site was used, but it looks like the same data from the other sites was used twice, first to construct a dispersal kernel, then to use that kernel to predict seed movement. This strikes me as an important source of error on the distance estimates that needs to be recognized by the authors, and perhaps means that the dispersal estimates for different individual birds are not at all independent. In any case, movements of birds living together cannot strictly speaking be considered as independent (as they are in your statistical models), as they will obviously influence each other e.g. through antagonistic behaviour or conspecific attraction.

I think the paper would be much stronger if it focused more on the movement data and included the third site (no campsite) data and the juvenile data, making a clear comparison in movement patterns between them all. The seed dispersal estimates are rough approximations (for the reasons given above), and should be a secondary part of the paper. In my opinion, they are not strong enough to make a stand alone paper in a high quality journal. Perhaps there is some plant to split the data into different papers, but that can be counterproductive.

Another thing of concern is the manner in which the paper is written as if seed dispersal only occurs through frugivory, ignoring the important dispersal of other seeds by rails through granivory, e.g. see Bartel et al. 2018, Thorsen et al. 2009, Green et al. 2016. I was left wondering what other plants are dispersed by these rails, e.g. given the importance of *Porphyrio* as a vector for *Eleocharis* (Bell 2000).

MINOR COMMENTS

28 overstated, e.g. there are many papers about the influence of disturbance on waterfowl, which are major seed vectors (e.g. papers by Madsen and Stillman)

62 “are reduced” should be “can be reduced” since this is not a universal finding

104 some explanation for extraordinary retention times should be offered, and these unpublished data should be made available. These are not so extraordinary in the light of Proctor 1968.

123 again this looks like the authors are ignoring the existence of seed dispersal by herbivorous and granivorous birds e.g. by introduced black swans (Green et al. 2008).

153 Exactly how these data were used needs explaining somewhere.

272 overstated, you present no information about dispersal quality since there are no results about how many virtual seeds would be deposited under the canopy of different trees, or in more suitable microhabitats. Hence you have not demonstrated highly effective dispersal. For that, you would at least need to present information on the estimated proportions of different habitats to which seeds are moved, and some idea of the value of each for seedling establishment.

333 in particular, it should be mentioned that dispersal can increase dispersal distances, e.g. when waterbirds are regularly disturbed and forced to fly to different wetlands.

342-44 presumably you mean 26 fleshy fruited species, I bet they disperse even more species that don't have a fleshy fruit (including those with a dry fruit).

482 This reference 27 seems incomplete, and ref 26 also seems wrong. Grey literature that is not available to a reader is best left uncited, or details of the important content provided in supplementary material.

REFERENCES CITED

- Bartel, R. D., J. L. Sheppard, A. Lovas-Kiss, and A. J. Green. 2018. Endozoochory by mallard in New Zealand: what seeds are dispersed and how far? *PeerJ* **6**.
- Bell, D. M. 2000. The ecology of coexisting *Eleocharis* species. PhD. University of New England, Armidale.
- Green, A. J., K. M. Jenkins, D. Bell, P. J. Morris, and R. T. Kingsford. 2008. The potential role of waterbirds in dispersing invertebrates and plants in arid Australia. *Freshwater Biology* **53**:380-392.
- Green, A.J., Brochet, A.L., Kleyheeg, E., Soons, M.B. 2016. Dispersal of plants by waterbirds. Pp 147-195 In: *Why birds matter: Avian Ecological Function and Ecosystem Services*. Eds. C.H. Şekercioğlu, D.G. Wenny, C.J. Whelan. University of Chicago Press.
http://digital.csic.es/bitstream/10261/146578/1/why-birds-matter_ch6_labelled.pdf

- Madsen, J., and A. D. Fox. 1995. Impacts of hunting disturbance on waterbirds - a review. *Wildlife Biology* **1**:193-207.
- Proctor, V. W. 1968. Long-distance dispersal of seeds by retention in digestive tract of birds. *Science* **160**:321-322.
- Stillman, R. A., A. D. West, R. W. G. Caldow, and S. E. A. L. D. Durell. 2007. Predicting the effect of disturbance on coastal birds. *Ibis* **149**:73-81.
- Thorsen, M. J., K. J. M. Dickinson, and P. J. Seddon. 2009. Seed dispersal systems in the New Zealand flora. *Perspectives in Plant Ecology Evolution and Systematics* **11**:285-309.

Appendix B

1st July 2019

Dear Editor,

We thank the reviewers for their constructive comments and welcome the opportunity to resubmit our revised manuscript. As suggested by the three reviewers, we have added the dataset, methods, and results of the seed retention time experiment into this work, which has significantly strengthened the paper. This makes the basis for our mechanistic model much clearer. We have also made the other small changes suggested by reviewers, and have responded to their specific comments below in green type.

Thank you for your consideration,

Dr. Jo Carpenter

Associate Editor's comments:

The reviewers of your paper are broadly positive towards it, though each indicate a number of modifications are necessary before it may be considered for publication. Importantly, it has been observed that only a portion of your data have been made available to reviewers (and thus also to readers in the event the paper is published). In line with the journal's data access policy, you should make the dataset(s), code, or other digital research materials available at submission - we will not be able to accept the paper unless you can not only respond effectively to the reviewers' concerns but also provide the additional (and currently absent) data. Please contact the editorial office with any queries you may have before you resubmit.

Comments to Author:

Reviewers' Comments to Author:

Reviewer: 1

Comments to the Author(s)

Manuscript ID: RSOS-190397

GENERAL COMMENTS:

The manuscript entitled 'Long seed dispersal distances by an inquisitive flightless rail (*Gallirallus australis*) are reduced by interaction with humans' is an original work through which authors combine GPS data and gut retention time to estimate seed dispersal distances provided by the New Zealand weka, and evaluate the impact that a strong contact with humans can have on this service. The study has involved a major field effort. It provides interesting results on cryptic function loss and opens debate on conservation strategies. The manuscript is clearly written, easy to follow and concise. However, I have some comments that I hope they help to improve the manuscript.

Thank you for your helpful comments. This study did have a major field effort and we appreciate the reviewer acknowledging that.

-ABSTRACT:

l. 38-40: I think that reporting the number of individuals analysed is not relevant in the abstract. Maybe you could rewrite this part of the abstract: 'However, we found evidence of a significant human impact on the seed dispersal distances of weka, birds occupying areas of high human use showing 34.8-40.9% shorter distances than their more remote counterparts'.

We have made these changes as suggested. It now reads: "However, we found evidence of a significant human impact on the seed dispersal distances of weka, with birds occupying areas of high human use performing 34.8-40.9% shorter distances than their more remote counterparts." Lines 38-41.

- KEYWORDS: I suggest replacing 'habitat change' by 'habitat use', which I think fits better to the studied situation in which structural anthropogenic modifications of the landscape are not described.

Done.

- INTRODUCTION:

l. 54: add a comma after 'noise'

We are actually referring to noise pollution specifically, not noise and pollution, so a comma isn't appropriate here.

l. 61-64: your focal seed disperser species is a bird, so why don't you change this example by highlighting what has been reported in birds (ref. 6)?

We searched for possible bird examples but thought that the cited mammal paper is more relevant to this case.

l. 71. This is a very general statement about seed dispersal consequences, so I think that ref. 9 is not necessary here (it refers to a certain plant species). But if you want to keep it, it should be necessary to fix it in the reference list (author names are not correctly written and two of them are repeated).

We have removed ref 9 as suggested.

l. 96. I suggest adding the authorship of the species the first time you mention it in the manuscript. The same for other taxa.

We have added the authorship for weka, it now reads "In New Zealand, one species that has a high level of human interaction is the weka (*Gallirallus australis*; Sparrman, 1786), an inquisitive flightless rail." Line 96.

l. 104. Carpenter 2018 unpubl. data: Why don't you include these data on gut retention time as part of this study?

This is a good point. We originally were planning to publish the seed retention data separately, but we agree with the reviewer that it should be part of this manuscript. We have added the seed retention time methods and results to this paper, and have extended the discussion to accommodate the new data.

I. 110. ...to study weka seed dispersal distances.

Amended, the sentence now reads “We used a mechanistic model approach to study weka seed dispersal distances.” Line 109.

METHODS

I. 124. birds (plural)

Corrected, thank you.

I. 130-138: Any reference supporting these descriptions of the plant species?

We have added a reference to support the plant species descriptions (The Flora of New Zealand, Vol. 1).

“Miro is a tree that grows to 25 m tall and occurs throughout New Zealand. Its fruits are 12-15 mm in diameter, with a fleshy exocarp and a hard, woody seedcoat 1.5-2 mm thick that encases the single seed [27]. Hinau trees grow up to 20 m tall and occur in lowland conifer-broadleaf forest throughout the North Island and the northern South Island. Its fruits average 9.2 mm diameter, with a carbohydrate-rich exocarp and mesocarp and a hard woody seedcoat protecting the single seed [27].” Lines 134-140.

I. 139 onwards: The study has an important spatial component, so It would be nice to see a figure with a map showing the three sites studied and the area occupied by the campsites at the Mahinapua and Goldsborough Reserves. Besides, the current manuscript has only one figure, so maybe a first figure showing graphically the experimental design could be part of the body of the ms.

We have added a map showing the location of the three study sites in New Zealand (to give overall context), and then two satellite images of Mahinapua and Goldsborough to show the area occupied by campsites (now Figure 2).

I. 166. Justify why the device shouldn't be heavier than 5% of the bird's body mass and add a reference (e.g. Geen et al 2019. Effects of tracking devices on individual birds – a review of the evidence. Journal of Avian Biology, e01823)

Done, the sentence now reads: “Birds were only fitted with a GPS and harness if the combined unit was no more than 5% of their body weight, as heavier devices relative to bird body mass can result in negative effects on foraging, locomotion, and physiology [31].” Lines 225-228.

I. 167. Why 14 days of monitoring? I guess because of the lifespan of the GPS batteries, but justify why.

We have added the following clarification: “After 14 days (the battery life of the GPS), we recaptured the weka and removed the transmitter and harness.” Lines 229-233.

I. 170. Were these three birds those from Mahinapua Reserve?

Two were from Mahinapua, one was from Goldsbrough. We have added that detail to the text: “Three weka were unable to be recaptured after the two week monitoring period (two individuals from Lake Mahinapua and one individual from Goldsbrough), and four weka had their GPS devices fail.” Lines 233-235.

I. 173. Report n of males and females, and n of juvenile and adults.

Done, the sentence now reads: “This left us with GPS data from 39 weka (Lake Mahinapua n = 13, Goldsbrough n = 12, Ulva Island n = 14), including both sexes (F = 11, M = 19, 5 individuals from Ulva Island and 4 juveniles from Goldsbrough and Mahinapua could not be sexed) and juvenile (n = 4) and adult individuals (n = 35).” Lines 235-239.

I. 175. Report the % of GPS fixes excluded.

We have included the following sentence: “The final dataset retained 66% of the original fixes.” Line 246.

I. 183 onwards (Mechanistic model). As required by the journal: “Datasets, code, and other digital materials should be deposited in an appropriate, recognised, publicly available repository”. Therefore I recommend you to make the R-code for the Mechanistic model available as supporting information.

This is a good point, the code has been added as an ESM (ESM 3).

I. 189. As pointed above, I think that this study focused on the estimation of seed dispersal distances by combining GPS data and information about gut retention time could be the place where properly report the data you have on seed retention time.

We agree, see above comment.

I. 196. Delete ‘only’

Done.

I. 218-225. It is not clear from the text if n=10 camp followers includes the juveniles or not, because they are also camp followers according to their home range. Rewrite this part in order to clarify that juvenile birds are n=5 and the final analysis was performed with 21 adult birds.

Thank you for pointing this out. We have attempted to clarify this and the text now reads: “Individuals whose core home range (defined as the 70% isopleth [29]) overlapped with the campground were defined as birds that had a high level of human interaction (hereafter referred to as ‘camp followers’). Birds at the two sites whose core home range did not overlap with a campground were categorised as ‘remote’ birds. We only used adult birds for the analysis as juvenile weka have different movement patterns to adult birds [35] and all the juvenile birds we captured were camp followers (n = 4), which would have confounded the model. Our analyses were therefore performed on 10 adult camp follower birds, and 11 adult remote birds.” Lines 284-292.

I. 235. You included 'site' as a random effect in your linear mixed effect model. However, you are only considering 2 sites, and the use of random factors with less than 5-6 levels is not recommended. Crawley [p. 670: Crawley, M. J. 2002. Statistical Computing: An Introduction to Data Analysis using S-PLUS] recommended treating variables as fixed factors when there aren't enough levels of a factor.

As published by Bolker et al. in the following blog:

<http://bbolker.github.io/mixedmodels-misc/glmmFAQ.html#should-i-treat-factor-xxx-as-fixed-or-random>

Are there enough levels of the factor in the data on which to base an estimate of the variance of the population of effects? No, means [you should probably treat the variable as] fixed effects.

This is a good point. We have rerun the model with site as a fixed effect rather than a random effect. "Median dispersal distance for each bird for each plant species (obtained from the mechanistic model) was the response variable, while human interaction (or not), site, and plant species were the fixed effects." Lines 294-297. Also see Table 2.

- RESULTS

I. 243. This info (< 7%) is redundant. You already said that 93-96% of the seeds are dispersed away from the parent tree.

Noted, we have removed the latter part of the sentence. It now reads: "The mechanistic model estimated that weka dispersed 93-96% of seeds away from the parent tree (assuming a canopy radius of 10 m; Table 1)." Lines 325-326.

I. 253. Cite Table 2 instead of Table 1.

Done, thank you.

DISCUSSION

I. 268 You cite an important study on spatial patterns of seed dispersal, but I miss here other important and more recent studies reporting seed dispersal kernels by birds of different sizes to which it would be interesting compare and discuss your dispersal kernel:

e.g.

Jordano et al. 2007. Differential contribution of frugivores to complex seed dispersal patterns. PNAS, 104

González-Varo et al. 2017. Unravelling seed dispersal through fragmented landscapes: Frugivore species operate unevenly as mobile links. Molecular Ecology, DOI: 10.1111/mec.14181

Thank you for highlighting those references. We have added text comparing the weka dispersal kernels with other studies. "The weka dispersal kernels were similar to seed dispersal kernels calculated for medium sized volant birds dispersing *Prunus mahaleb* in Spain [50], and for *Sturnus unicolor* and *Turdus philomelos* dispersing *Olea europaea* var. *sylvestris* [51]." Lines 394-397.

I. 272. I would be more careful when talking about effectiveness in seed dispersal (also in I.

345), since it involves many other drivers besides seed dispersal distance. I think that your results are very important for the qualitative component of the seed dispersal effectiveness. I also but think that you can talk about effectiveness when discussing your results together with previous information on the quantitative component of seed dispersal.

This is a good point. The sentence now reads: "Since seeds that are deposited beneath parent canopies can suffer from disproportionate mortality due to density- and distant-dependent mortality [52], this result demonstrates that weka provide good qualitative seed dispersal [53]". Lines 399-402.

We have also amended the statement on line 483 to talk about high quality seed dispersal rather than effective seed dispersal.

I. 284-286. But are these other frugivores capable of dispersing the seeds of the large-fruited Hinau and Miro? Please, rewrite to clarify that weka and these other frugivores overlap in their fruit diet so they could provide complementary seed dispersal services.

We have clarified this point and the text now reads: "Despite being flightless, the mean dispersal distances of weka compare favourably to dispersal distances calculated for some of New Zealand's volant frugivores, which are also capable of consuming hinau or miro (although it should be noted that two common frugivores, bellbirds (*Anthornis melanura*) and tui (*Prosthemadera novaeseelandiae*) have not been recorded consuming hinau)." Lines 406-410.

I. 292. Do you know if juveniles perform dispersal movements in a certain time of the year? If so, could these movements involve long distance dispersal events for certain plant species with overlapping fruit phenologies? (This is just a question that rises to me when reading this line).

Juvenile weka have been recording dispersing large distances from their natal home ranges by Coleman et al. 1983 (Notornis), and we reference this in the text in terms of its potential for long distance dispersal events. Bramley 2001 (Notornis) studied juvenile dispersal of weka also, but dispersal distances were generally small (ranging from no dispersal to 3.5 km), although most would have still resulted in long distance dispersal events.

I. 304-305. Not only on enhancing gene-flow but also for colonising new sites and tracking the climatic envelope.

Very true, we have amended the text to include these. It now reads: "These long distance dispersal events could profoundly enhance genetic flow across extensive landscape scales, as well as helping plants track their climatic envelope and colonize new sites." Lines 435-437.

I. 317. Add references to support this.

We have added the following reference to support the statement:

Sebbenn AM, Carvalho ACM, Freitas MLM, Moraes SMB, Gaino APSC, Da Silva JM, Jolivet C, Moraes MLT. 2011 Low levels of realized seed and pollen gene flow and strong spatial genetic structure in a small, isolated and fragmented population of the tropical tree *Copaifera langsdorffii* Desf. *Heredity*. **106**, 134–145.

Reviewer: 2

Comments to the Author(s)

See the attached file

Reviewer: 3

Comments to the Author(s)

Carpenter et al. explore the efficiency of flightless birds, in particular the New Zealand rail weka, as seed dispersers, and further investigate the effects of increased exposure to human presence in the landscape. Using GPS tracks from some 40 birds in three different sites combined with mechanistic modeling of seed retention time, they find that while volant birds are usually presumed more efficient seed dispersers, the flightless weka is a surprisingly efficient disperser due to relatively large home ranges and unusually long seed retention time. However, wekas are curious birds that are attracted to human activity, and the authors noticed reduced mobility among rails in areas with human presence (e.g. campgrounds), leading to cryptic loss of ecosystem services through reduced seed dispersal.

This is overall a neat and well-designed study that delivers both counterintuitive and important results. I have unusually few remarks, but I do find one issue generally problematic: The mechanistic model of seed retention is based on an unpublished dataset of seed retention times by the main author (“Carpenter 2018 unpubl. data”), from which the statement also stems that the weka has the longest seed retention time in the world. Since this is one of the two major datasets that the present paper is based upon, I do not think it is acceptable to only refer to this as an unpublished dataset (and not make it available). In the present paper, the probability distributions of seed dispersal distances are presented, but as reader I am only presented with a couple of descriptive statistics of the seed retention time. Perhaps this dataset is meant to be published in a separate manuscript, but if that manuscript is not accepted, it renders the present one premature. In order to avoid that, I would argue that the authors should include methods and results for the collection of that dataset too in this manuscript. I also note that the authors have answered “Yes” to the question whether new data is presented (true), and state that “All data used in this paper are available as ESM 1”. The GPS logging dataset in ESM 1 is extensive and well curated, but it only covers the tracking, and the latter statement is thus incorrect. Thus, in order to adhere to RSOS standards, the seed retention data should make up ESM 2, and be presented in the manuscript.

Thank you for the overall positive response to the MS. The reviewer has a very good point about the unpublished dataset of seed retention times. We have rectified this problem as stated above. The seed retention time dataset is available as ESM2.

I don't think that the opening statement is correct. The authors claim that “Humans have modified approximately 50 to 70% of the earth's surface” and refer to Barnosky et al. (2012) Nature. However, I do not believe that these numbers are supported by Barnosky et al. (especially look at terrestrial vs. marine parts of the Earth, and current vs. projected transformation).

Thank you for pointing this out. We have changed the sentence and the reference. It now reads: “Humans have modified over half of the earth's terrestrial surface [1], with profound consequences for the species that use those habitats.”

1. Hooke RLB, Martín-Duque JF, Pedraza J. 2012 Land transformation by humans: A review. *GSA Today* **22**, 4–10. (doi:10.1130/GSAT151A.1)

The authors consistently use “weka” as plural form. As far as I, as non-newzeelander, understand, “wekas” would be the correct plural form of weka, although I also understand that “weka” is sometimes used also as plural.

We can see why this might be confusing. Weka is a te reo word from the maori (indigenous people of New Zealand) language. There is no ‘s’ in te reo so ‘weka’ can be both singular and plural. We have added this to the text and it now reads: “Weka (as a maori word, the spelling is the same for singular and plural) are bold, opportunistic birds that frequently aggregate at areas of high human use, such as campsites or picnic areas [19]”. Line 97.

References 5, 24, 26, 38, 42 have missing or incorrect details or incorrect formatting.

We have corrected these references.

As a very minor comment, I would advise the authors to present Figure 2 as boxplots rather than a bar chart, given the type of data.

We appreciate this suggestion and have changed the graph to show boxplots rather than bars (new Figure 5).

Apart from the treatment of seed retention data, all other comments are minor, and would render an “Accept with minor revision”. In my opinion, if the availability and treatment of seed retention data (preferably by inclusion in the manuscript and as supplement or data deposition), this manuscript is in good shape and qualifies for publication in RSOS. However, as is, I therefore recommend a “Major revision”.

This is a study of estimated seed dispersal distances by populations of a flightless rail in three sites. From my understanding, adults and juveniles are tracked at three different sites. Then seed dispersal was estimated only for adults, and only for two of the sites (those with some adult birds using a campsite and other birds using other habitats). Seed dispersal is estimated based on unpublished retention time data, and it seems that neither these data nor the movement data per se are made available by the authors (Q: is that not against the policy of this journal?).

Thank you for your comments. Seed dispersal distances were calculated for all birds from all sites. We then just used adult birds at two sites with a hub of high human use to test whether seed dispersal distances were shorter when birds had more human interaction. The movement data are fully accessible as ESM1, and the seed retention time methods, results, and data are now also available.

There seems to be no use of real data on true diet or the location or the timing of seed

ingestion data, so in that sense the seed dispersal distance estimates cannot be considered to be reliable. Neither are there data presented that show that the rails disperse these plants at these sites, although perhaps some such data are included in reference 21.

Mechanistic models of seed dispersal kernels using movement data and seed retention time data are frequently used to estimate seed dispersal distances for animals with no use of diet data or timing of seed ingestion (e.g. Holbrook and Smith 2000, Santamaria et al. 2007, Ward and Paton 2007, Lenz et al. 2011). Other diet studies have already demonstrated that weka eat many fruits (e.g. Carpenter et al. 2018, Coleman et al. 1983) and the exact timing of ingestion is irrelevant as weka can be active and feed at all hours of the day and night (most other studies simply model their seed ingestion times during the day for diurnal species). We did not collect data to show that weka specifically consume hinau and miro at these sites as weka have already been shown to eat these two plant species' fruits (Carpenter et al. 2018, Kelly et al. 2010).

The reported effects seem believable and are somewhat equivalent to showing that ducks in urban ponds, where people feed them bread, fly around less (and disperse seeds over shorter distances) than ducks in natural wetlands. This is not very surprising, but there is only a weakly significant effect and the statistical analyses do not seem to me to be robust, since there is really only a sample size of two (i.e. the two locations where birds are compared with and without a campsite).

We respectfully disagree with the reviewer. Each individual bird gives a data point (median seed dispersal distance) and therefore the sample size for the analysis is 21 (21 birds). Birds could behave differently so are independent tests of the hypothesis. Moreover, we used two sites, each with a treatment (human interaction) and non treatment (remote) area. This spatial replication covering birds in two different areas makes our analyses even more robust.

Furthermore, it may be that there is some other confounding variable, e.g. the campsites are positioned in low-lying areas with more emergent vegetation that provides more natural food (if these rails behave anything like Porphyrio).

We never saw weka foraging for natural food in the campsites, which consist largely of mown grass. The nearby non-campsite areas are similar in altitude, so we think there are no other obvious confounding factors.

I did not understand how the movement data from the third site was used, but it looks like the same data from the other sites was used twice, first to construct a dispersal kernel, then to use that kernel to predict seed movement. This strikes me as an important source of error on the distance estimates that needs to be recognized by the authors, and perhaps means that the dispersal estimates for different individual birds are not at all independent.

We are unsure what the reviewer means here. Movement data from weka at all three sites was used in a mechanistic model to create seed dispersal kernels for weka. This is the common approach

when using movement data and seed retention time data in a mechanistic model to create seed dispersal kernels.

In any case,

movements of birds living together cannot strictly speaking be considered as independent (as they are in your statistical models), as they will obviously influence each other e.g. through antagonistic behaviour or conspecific attraction.

We agree that birds that occur in the same habitat may influence each others movement. However, as this is environmentally realistic (i.e., these birds would almost never occur in complete isolation) it leads to more accurate estimates of how far these birds actually disperse seeds in a 'normal' ecological setting. All the papers we are aware of that compute mechanistic models for species' seed dispersal kernels use animals that are part of the same community and therefore may be influencing each others' movements (e.g. Lenz et al. 2011, Wotton and Kelly 2012).

I think the paper would be much stronger if it focused more on the movement data and included the third site (no campsite) data and the juvenile data, making a clear comparison in movement patterns between them all. The seed dispersal estimates are rough approximations (for the reasons given above), and should be a secondary part of the paper. In my opinion, they are not strong enough to make a stand alone paper in a high quality journal. Perhaps there is some plan to split the data into different papers, but that can be counterproductive.

As we mention above, we believe our method of constructing mechanistic models to estimate seed dispersal distances from the movement data and the seed retention data is a tried and tested approach that has been used to estimate seed dispersal kernels for multiple species with good results. We actually think the seed dispersal distances we calculated for weka as a species are particularly robust because we used such a large sample size of birds ($n = 39$) across three sites. Most studies to date have smaller sample sizes of birds and they only study their movement at one site (e.g. Holbrook and Smith 2000; Lenz et al. 2011).

Another thing of concern is the manner in which the paper is written as if seed dispersal only occurs through frugivory, ignoring the important dispersal of other seeds by rails through granivory, e.g. see Bartel et al. 2018, Thorsen et al. 2009, Green et al. 2016. I was left wondering what other plants are dispersed by these rails, e.g. given the importance of *Porphyrio* as a vector for *Eleocharis* (Bell 2000).

This is a good point, and we appreciate the reviewer highlighting this for us. We have added text to acknowledge that weka could also disperse seeds through granivory, and that other rails have been reporting doing so. See below.

"Weka also may disperse the seeds of monocots or dry seeds through granivory, an overlooked yet important seed dispersal mechanism that has been demonstrated for other rail species such as Eurasian Coots (*Fulica atra*)[62]." Lines 480-483.

“More broadly, our research into the seed dispersal capabilities of weka suggests that other rails across the Pacific may have been important seed dispersers, even though they are rarely mentioned as such. However, *Porphyrio* species have been recorded as significant vectors of *Coprosma* seeds both within and between New Zealand and the Pacific, and *Scleria* seeds were found in the gut of *Porphyrio* in Fiji (cited in [63]).” Lines 499-502.

MINOR COMMENTS

28 overstated, e.g. there are many papers about the influence of disturbance on waterfowl, which are major seed vectors (e.g. papers by Madsen and Stillman)

This is a good point. We have modified the sentence and it now states: “Human presence is becoming increasingly ubiquitous, but the influence this has on the seed dispersal services performed by frugivorous animals is largely unknown”. Lines 27-28. We feel this is a fair statement as while many studies have examined how disturbance affects animals (some of which are seed dispersers), very few have looked specifically at how seed dispersal services were affected as a result.

62 “are reduced” should be “can be reduced” since this is not a universal finding

True, we have modified the sentence as suggested and it now reads: “Due to these mechanisms, movements of mammals in intensively human-modified areas can be reduced by half to two-thirds compared with individuals in areas with less human modification [5], and similar reductions in vagility have been documented for other taxa (e.g.[6,7]).” Lines 60-64.

104 some explanation for extraordinary retention times should be offered, and these unpublished data should be made available. These are not so extraordinary in the light of Proctor 1968.

We have now included the methods, results, and data from the seed retention time study in this paper. We have also inserted a paragraph in the discussion which compares weka seed retention times to those of other birds and demonstrates that they are some of the longest avian seed retention times ever recorded.

123 again this looks like the authors are ignoring the existence of seed dispersal by herbivorous and granivorous birds e.g. by introduced black swans (Green et al. 2008).

This is a good point, we have changed the sentence and it now states: “Weka are one of the largest (mean 900 g, range 400-1700 g) extant frugivorous birds in New Zealand [24].” Lines 125-126.

153 Exactly how these data were used needs explaining somewhere.

It is explained in the latter half of the paragraph: “Ulva Island had no hub of high human use so was not used for this analysis, but was still used to estimate overall weka seed dispersal distances.” Line 216-217.

272 overstated, you present no information about dispersal quality since there are no results about how many virtual seeds would be deposited under the canopy of different trees, or in more suitable microhabitats. Hence you have not demonstrated highly effective dispersal. For

that, you would at least need to present information on the estimated proportions of different habitats to which seeds are moved, and some idea of the value of each for seedling establishment.

We see the reviewer's point that data on microsite deposition would be excellent to have, and this is an important component of dispersal quality. However the distance that seeds are dispersed and the proportion of seeds that are dispersed away from the parent canopy are still important facets of seed dispersal quality, so we do not think the sentence is overstated.

333 in particular, it should be mentioned that dispersal can increase dispersal distances, e.g. when waterbirds are regularly disturbed and forced to fly to different wetlands.

This is a very good point. We have added the following sentence to acknowledge this: "However, it should be noted that in some cases human disturbance may have the potential to increase seed dispersal distances of some species, for example when water birds are regularly disturbed and fly to different wetlands [58]." Lines 449-452.

342-44 presumably you mean 26 fleshy fruited species, I bet they disperse even more species that don't have a fleshy fruit (including those with a dry fruit).

That is true, we have amended the sentence and it now reads: "They have been recorded consuming the fruits of over 26 native fleshy fruited plant species, including some of New Zealand's largest-seeded species [61], and the fruits of low-growing divaricating shrubs. Weka also may disperse the seeds of monocots or dry seeds through granivory, an overlooked yet important seed dispersal mechanism that has been demonstrated for other rail species such as Eurasian Coots (*Fulica atra*)[62]." Lines 477-483.

482 This reference 27 seems incomplete, and ref 26 also seems wrong. Grey literature that is not available to a reader is best left uncited, or details of the important content provided in supplementary material.

We have fixed the references.

REFERENCES CITED

- Bartel, R. D., J. L. Sheppard, A. Lovas-Kiss, and A. J. Green. 2018. Endozoochory by mallard in New Zealand: what seeds are dispersed and how far? PeerJ 6.
- Bell, D. M. 2000. The ecology of coexisting *Eleocharis* species. PhD. University of New England, Armidale.
- Green, A. J., K. M. Jenkins, D. Bell, P. J. Morris, and R. T. Kingsford. 2008. The potential role of waterbirds in dispersing invertebrates and plants in arid Australia. *Freshwater Biology* 53:380-392.

Green, A.J., Brochet, A.L., Kleyheeg, E., Soons, M.B. 2016. Dispersal of plants by waterbirds. Pp 147-195 In: *Why birds matter: Avian Ecological Function and Ecosystem Services*. Eds. C.H. Şekercioğlu, D.G. Wenny, C.J. Whelan. University of Chicago Press.
http://digital.csic.es/bitstream/10261/146578/1/why-birds-matter_ch6_labelled.pdf

Madsen, J., and A. D. Fox. 1995. Impacts of hunting disturbance on waterbirds - a review. *Wildlife Biology* 1:193-207.

Proctor, V. W. 1968. Long-distance dispersal of seeds by retention in digestive tract of birds. *Science* 160:321-322.

Stillman, R. A., A. D. West, R. W. G. Caldow, and S. E. A. L. D. Durell. 2007. Predicting the effect of disturbance on coastal birds. *Ibis* 149:73-81.

Thorsen, M. J., K. J. M. Dickinson, and P. J. Seddon. 2009. Seed dispersal systems in the New Zealand flora. *Perspectives in Plant Ecology Evolution and Systematics* 11:285-309.

Appendix C

Reviewer comments to Author:

Reviewer: 2

Comments to the Author(s)

The paper has clearly improved a lot after full disclosure of the retention time experiment. However, I stand by my earlier comments which the authors have chosen to dismiss. In particular in two sites, birds are compared between two groups, those associated with a camp site and those not. In principle, there could be many reasons why there is a difference in movement between birds using one part of a habitat and another. There are only two places where there is a comparison made between camp and other. Hence it is perfectly possible that the results have a different explanation, the real sample size of importance is $N = 2$ study sites, in both of which a similar spatial pattern is recorded. Given this reality, and in the absence of other data I think the conclusions need to be presented with some more caution.

We thank the reviewer for their comments on the paper, which have greatly improved it. As the reviewer notes, however, we disagree about interpretation of the movement data. The key point is about the relevant level of independence for the question being asked. Within each area, we had multiple birds, varying in size, sex, experience, whether their ranges include the camp, and other attributes. Our stats use birds as replicates, because each bird makes its own decisions about movement and hence is an independent sample of movement data. When classified into two groups by one attribute (use camp areas vs not using camps), there was a significant difference in seed dispersal distance. The main reason for including two different sites is that the difference in bird movement associated with camp use at one site may be caused by some other unknown variable which happened to be confounded with camp location. By having two independent sites, and finding the same trend for birds when classified by camp, we gain confidence that the demonstrated difference in ranges of individual birds was causally linked to use of camp areas (and consequent interaction with humans), rather than caused by an unknown confounding factor. So the two sites are important for generalization about causes, but the relevant sample size for testing numerical differences in bird range areas remains the number of birds. Having said that, two independent sites make unknown confounding factors unlikely, but not impossible. Hence in recognition of the reviewer's point, we added to the discussion: "In addition, it is possible that there were other unmeasured factors associated with the campsites (other than human presence) which affected weka seed dispersal distances."

In this new version I have the following specific comments:

There is a lack of clarity about what is meant by "quality of seed dispersal" e.g. in line 105 or 483. If we compare the paper with Schupp et al. 2017, I do not see what component of quality (as defined in that paper) is being studied herein. The effectiveness framework as presented says very little about dispersal distance, unlike this rail paper. I suggest it is best to just talk of dispersal distance rather than "quality".

We have changed all references in the paper to 'dispersal quality' to 'dispersal distance', as recommended by the reviewer.

Line 113 Explain/spell out what RFID is.

We have added more detail to the sentence. It now reads: “Our aims were to: 1) estimate weka seed retention times using a novel RFID (radio-frequency identification) microchip method, which records when ingested microchipped seeds were still inside weka.”

132-134 A clear justification is required here for why you are modelling dispersal of these two plant species, since you present no data on the consumption or abundance of these two plant species in your study area. Ultimately it is because you have data from a captive experiment, but this needs pointing out. And some idea of the distribution of these two plant species in the study area should be given.

We have added the following text to clarify why we were interested in these two plant species in particular: “We were particularly interested in these two species as weka have been shown to be the most important disperser for hinau in terms of fruit removal [20], and they are also likely to be an important disperser for miro where weka they are present as its large fruits are consumed by few other species [21]. In addition, both plant species feature some fruit traits (e.g. early abscission, thick seed coats) that suggest they may be partially adapted for dispersal by flightless birds such as weka [26], and are found throughout all (miro) or nearly all (hinau) of the range of weka.”

At the three field sites used for GPS tracking, both plants are present except for hinau being absent at Ulva Island. We thought it would be potentially confusing to detail this (in a later part of the Methods section), as the field sites only measured bird movements, not ingestion of particular plant species; the actual plant species growing at the field sites were irrelevant to those experiments. If a reader was interested in this, the above new text in combination with Fig 2 would make clear to them that the Ulva site was beyond the range of hinau.

160-161 Discuss the possible consequences of removing the fleshy exocarp from miro seeds on the results, especially retention time. You assume that removing pulp makes no difference, but what studies in other organisms support such an assumption? Furthermore, the seeds were smeared with cheese, might that not influence retention time? At least say why you think not.

This is a good point. We have added the following text to the discussion: “In addition, our removal of the fruit pulp of miro may have affected weka seed retention times for that species, as the secondary metabolites within fruit pulp can alter seed retention times [47–49]. Here we found long retention times both in hinau (which had the fruit pulp intact) and miro (which had the fruit pulp removed), although we cannot determine what might have happened if fruit pulp had been removed for hinau or left intact for miro. Similarly, the cheese we smeared on the fruits may have affected the weka seed retention times, although we think this is unlikely given how long the seeds remained inside the birds.”

433 “very long distance dispersal” seems a significant exaggeration to me for a flightless bird. I would apply that term rather to birds that move a much greater distance of hundreds of km, e.g. Viana et al.

This is a fair point. We have removed the ‘very’ from the sentence so it now reads: “These occasional long distance movements, coupled with seed retention times that can reach 40 days, demonstrate that weka almost certainly generate rare long distance dispersal events

beyond what could be documented with our mechanistic model (which also underestimated dispersal distances for several technical reasons as noted in Methods)."

REFERENCES CITED

Schupp, E.W., Jordano, P., Gomez, J.M., 2017. A general framework for effectiveness concepts in mutualisms. *Ecology Letters* 20, 577-590.

Viana, D.S., Santamaria, L., Figuerola, J., 2016. Migratory Birds as Global Dispersal Vectors. *Trends in Ecology & Evolution* 31, 763-775.

Reviewer: 3

Comments to the Author(s)

I think that the authors have addressed all of the issues brought up by the reviewers in a satisfactory manner, and improved the manuscript greatly. With the explicit inclusion of the seed retention estimation, and the provision of that dataset, the manuscript is now "complete" and meets the open access standards of RSOS. With a minor edit (see below) I think that this study is now suitable for publication.

After reading the revised manuscript, I only find one minor issue that could easily be addressed. For some reason, in a single passage, only scientific names are used. I would strongly suggest that in the final version of the manuscript, the common names are inserted for lines 395–397. The birds are spotless starling (*Sturnus unicolor*) and song thrush (*Turdus philomelos*), the trees wild olive (*Olea europaea* var. *sylvestris*) and mahaleb cherry (*Prunus mahaleb*) [although the latter one may also be known also under other common names].

*We thank the reviewer for their suggestions. We have made the suggested amendment. The sentence now reads: "The weka dispersal kernels were similar to seed dispersal kernels calculated for medium sized volant birds dispersing mahaleb cherry (*Prunus mahaleb*) in Spain [51], and for spotless starling (*Sturnus unicolor*) and song thrush (*Turdus philomelos*) dispersing wild olive (*Olea europaea* var. *sylvestris*) [52]."*